# Evaluating the climate impact of aviation emission scenarios towards the Paris agreement including COVID-19 effects

Volker Grewe [1,2,3 ✉], Arvind Gangoli Rao [2,3], Tomas Grönstedt[3,4], Carlos Xisto [3,4], Florian Linke [3,5], Joris Melkert[2,3], Jan Middel[3,6], Barbara Ohlenforst [3,6], Simon Blakey [3,7,8], Simon Christie [3,9], Sigrun Matthes [1,3] & Katrin Dahlmann[1,3]

Aviation is an important contributor to the global economy, satisfying society's mobility needs. It contributes to climate change through $CO_2$ and non-$CO_2$ effects, including contrail-cirrus and ozone formation. There is currently significant interest in policies, regulations and research aiming to reduce aviation's climate impact. Here we model the effect of these measures on global warming and perform a bottom-up analysis of potential technical improvements, challenging the assumptions of the targets for the sector with a number of scenarios up to 2100. We show that although the emissions targets for aviation are in line with the overall goals of the Paris Agreement, there is a high likelihood that the climate impact of aviation will not meet these goals. Our assessment includes feasible technological advancements and the availability of sustainable aviation fuels. This conclusion is robust for several COVID-19 recovery scenarios, including changes in travel behaviour.

[1] Deutsches Zentrum für Luft- und Raumfahrt (DLR), Institut für Physik der Atmosphäre, Oberpfaffenhofen, Germany. [2] Faculty of Aerospace Engineering, Delft University of Technology, Delft, Netherlands. [3] ECATS International Association, Brussels, Belgium. [4] Mechanics and Maritime Sciences, Chalmers University of Technology, Gothenburg, Sweden. [5] Deutsches Zentrum für Luft- und Raumfahrt, Lufttransportsysteme, Hamburg, Germany. [6] Royal Netherlands Aerospace Centre (NLR), Amsterdam, Netherlands. [7] University of Birmingham, Mechanical Engineering, Birmingham, UK. [8] University of Sheffield, Mechanical Engineering, Low Carbon Combustion Centre, Sheffield, UK. [9] Department of Natural Sciences, Manchester Metropolitan University, Manchester, UK. ✉email: Volker.Grewe@dlr.de

                                          

Fuel efficiency of jet aircraft has been increasing right from the dawn of jet aviation in the late '50 s and early '60 s. This improvement cannot be attributed to one single source but has been achieved by a combination of factors such as improvements of the airframe aerodynamics, weight reductions due to better engineering, materials and manufacturing techniques, larger engines with a lower specific thrust, higher overall pressure ratios and component efficiencies, lighter structures and lighter on-board systems. Kharina and Rutherford[1] report an average reduction in fuel consumption per passenger-km at the global fleet level of 1.3% per year over the years 1960–2014. Without any further specific measures this reduction is expected to continue at a similar rate until 2037[2] in a business as usual scenario.

Air transport as a sector has been growing rapidly in most regions of the world. The total number of passengers transported annually passed 4 billion in 2017. The number of flights in all regions of the world has increased (Supplementary Fig. 1) and aircraft have on average greater seating capacity and are operated with a higher load factor (Supplementary Fig. 2). It is expected that air transport will continue to grow in the coming decades. Airbus[3] predicts in its Global Market Forecast continued annual growth of 4.4% in revenue passenger kilometre (RPK) for the next two decades. Boeing[4] expects in its Commercial Market Outlook an annual growth of 4.6%. The effects of the COVID-19 pandemic are expected to only have a temporary effect on this growth.

Without any measure the climate impact of aviation will continue to grow. Several measures, both political and technical, are in place or will be introduced in the near future. Via a number of scenarios, we analyse their effect on global warming and assess the effectiveness of these measures. Since many of these measures are set top-down we also want to assess the technical feasibility. Therefore, we have performed a bottom-up expert assessment on the feasibility of technical advances and their effect on climate change. We confront the two approaches with each other.

The profitability for the airlines is small. Their average net profit per passenger is <10 USD (Supplementary Fig. 3). Competition amongst airlines is fierce and therefore sensitive to airline costs differences. Fuel costs play an important role, which is of particular concern for the uptake of sustainable alternative fuels (SAF) that currently have a significantly higher cost than conventional fossil fuels. The COVID-19 pandemic has led to a large decrease in the number of flights and passenger load factors in 2020. In May 2020, the International Civil Aviation Organisation (ICAO) estimated a decrease of global total available seat kilometres of 94% in April 2020 compared to the 2019 baseline. However, they expect a recovery leading to an annual decrease in available seat kilometres of 45% to 63% for 2020[5], but assume growth will resume beyond 2020.

Approximately 5% of the current anthropogenic climate change is attributed to global aviation[6,7] and this number is expected to increase since aviation passenger transport is projected to grow by ~4% per year whilst other sectors continue to decarbonise. Aviation emits carbon dioxide ($CO_2$), water vapour ($H_2O$), nitrogen oxides ($NO_x$), sulphate aerosols, compounds from incomplete combustion (unburnt hydrocarbons, UHC) and particulates (soot). The emitted species are transported in the atmosphere and alter a wide range of atmospheric processes including the formation of contrail-cirrus and ozone and the depletion of methane[7–9].

The formation of persistent contrails-cirrus depends on aircraft and fuel parameters as well as atmospheric conditions, as the propensity of contrail formation is higher in the cold and saturated atmosphere[10–12]. Contrail-cirrus influence the incoming solar radiation and the outgoing infrared radiation emitted by the

Earth and its atmosphere. The net change, the radiative forcing (RF), is on average positive and hence contrail-cirrus act to warm the climate[13]. The emitted nitrogen oxides ($NO_x$) react with hydroxyl radicals ($HO_x$), which eventually form ozone and contribute to the depletion of methane in the atmosphere. Therefore, emissions of nitrogen oxides increase the ozone concentration and decrease the methane concentration (which itself leads to a reduction in ozone production and is called primary mode ozone, PMO). Ozone and methane are greenhouse gases and changes in their concentrations cause changes in the RF, which are in total positive, i.e. leading to warming[7,14,15]. The net direct impact of aerosol emissions on RF (soot: warming and sulphate: cooling) is small[7] and are not further regarded in this study, whereas the impact of soot emissions on contrail-cirrus properties are important[16] and considered in our calculations (see 'Methods'). An open question, which is currently under investigation is whether aerosol emissions significantly alter or even induce natural clouds, both low-level and cirrus clouds[17].

The Advisory Council for Aviation Research and Innovation in Europe (ACARE) has set targets for the reduction of emissions in its Flightpath 2050 document[18]. Among these targets is a reduction of 75% of $CO_2$ and 90% of $NO_x$ emission per passenger-km by 2050. The datum for these reductions is a typical new aircraft in the year 2000. These targets are set for the research, with intended outcomes to be realised at a technological readiness level (TRL, The European definition of TRLs range from 1 to 9, i.e. from 'basic principles observed' to 'actual system proven in operational environment') of 6.

The ICAO of the United Nations has agreed on a global market-based measure scheme to abate the growth of $CO_2$ emissions from international aviation. This scheme is the Carbon Offsetting and Reduction Scheme for International Aviation (CORSIA). According to this scheme, the post-2020 growth in the sector must be offset such that the net carbon emissions do no longer grow. They must either be reduced via more efficient aircraft and/or the use of SAF or must be compensated via offsets. CORSIA starts as a voluntary pilot scheme in 2021 and becomes mandatory, with some exceptions, in 2027 for all member states[19]. Aviation is a growing sector that has committed to reduce net $CO_2$ emissions and thus contributes to the international goals of limiting climate warming 'to well below 2.0° C above preindustrial levels and pursuing efforts to limit the temperature increase to 1.5 °C above preindustrial levels', as stated in the Paris Agreement[20]. The Paris Agreement does not set emission targets for specific sectors. Furthermore, international aviation and shipping are not included in the national contributions that countries have to make to comply with the agreement. However, we assume that the international aviation community will contribute to the goal of the Paris Agreement. We will investigate the effect of measures and policies on global warming and also assess their feasibility. Thereby we will not distinguish between domestic and international aviation but treat the sector as a whole. There are two bridges to cross between the emission goals set by ACARE and ICAO and the climate targets set by the Paris Agreement: First, how do the emission goals translate into near-surface temperature changes, i.e. climate change. Second, how large are the non-$CO_2$ effects?

Here we close these gaps and show that the emissions goals set by Flightpath 2050 very likely will stabilise aviation's climate impact, though the sector's contribution to global warming remains considerable. Contrarily, we find that ICAO's offsetting scheme, CORSIA, will surpass the climate target set to support the 1.5 °C goal between 2025 and 2064 with a 90% likelihood. In both cases non-$CO_2$ effects will have a considerable contribution to aviation's climate impact, however, they are currently not included in ICAO's goal of climate neutral growth and only partly

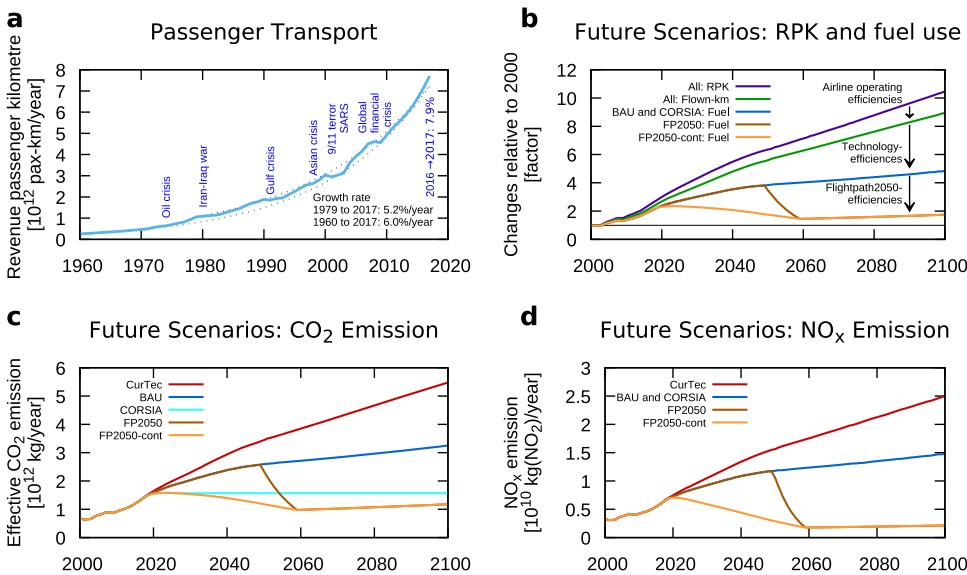

**Fig. 1 Global aviation transportation and related emissions for five scenarios.** They include the future use of current technology, i.e. without technology improvements (CurTec), with a business-as-usual future technological improvement (BAU), the offsetting scheme of the international civil aviation organisation (CORSIA), and 2 Flightpath 2050 scenarios which differ in the speed of technology improvements (FP2050 and FP2050-cont). **a** Revenue passenger kilometres as provided by ICAO; dotted lines provide exponential growth rates. **b** Future changes relative to their respective values in the year 2000 for revenue passenger kilometres (violet), flown distances (green), the fuel consumption of the scenarios BAU and CORSIA (blue), and the FP2050 scenarios (brown). **c** Future $CO_2$ emissions for the scenarios CurTec (red), BAU (blue), CORSIA (light blue), FP2050 with late technology advancements (dark brown) and continuous technology advancements (brown). Note that for CORSIA the effective $CO_2$ emission is considered, including reductions due to the use of sustainable alternative fuels (SAF) and capping net emissions. **d** as bottom-left, but for $NO_x$ emissions; note that the $NO_x$ emissions for BAU and CORSIA are identical. The order in the legend is the same as the lines appear in the graph.

addressed in Flightpath 2050. We assess the feasibility of achieving the Flightpath 2050 goals by technological improvements and the availability of sustainable alternative fuels as an ECATS (Environmentally Compatible Air Transportation System) expert group and reveal the risk of a large discrepancy, leading to an increasing aviation induced global warming effect rather than stabilisation.

## Results and discussion

**Top-down scenarios for future aviation.** Figure 1a presents the global growth of revenue passenger kilometres, showing an exponential increase between 5.2 and 6% per year (dotted lines). From this basis, we developed eight top-down scenarios, which consider a further increase in aviation, though with a decreasing rate of growth (down to 1.2%/year). These rates are based on simulations of the aviation sector, relying on the Randers scenario[21], which is independent from aircraft manufactures. This scenario was employed within the WeCare project[22] and considers worldwide saturation effects of economic growth. This Randers scenario leads to a growth rate of 1.2%/year in 2050 which we extrapolate to 0.8%/year in 2100 (see also Supplementary Material). Our industry-independent scenario shows lower estimates of the transportation volume for the coming two decades compared to the Airbus and Boeing forecasts (see above), though still slightly higher than other estimates for 2050[23,24]. Advances in airline operating efficiency, including changing the type of aircraft, the number of seats and load factor lead to a reduced increase of flown kilometres (Fig. 1b; green line) compared to the transport volume measured in RPKs (violet line). More fuel-efficient technologies even lead to a smaller increase in fuel use compared to flown distances (blue and orange lines). Taking the targets of Flightpath 2050 into account, a more aggressive reduction in emissions can be achieved up to 2050. In the scenario FP2050, we consider a development of these

technologies until 2050 followed by an introduction into the market. In the scenario FP2050-cont we apply a continuous introduction of these innovative technologies into the market (Fig. 1b, early and continuous/late introduction light/dark brown, respectively). Using these assumptions, the modelled results show that after 2050 the increase in RPK is balanced by technology enhancements leading to almost constant fuel consumption until 2100.

We take into account five different scenarios (Table 1): (1) Current Technology (CurTec), which describes the emission pathways with current (2012) technology, (2) Business-as-usual (BAU), which, in addition, takes into account some of the future improvements in technology, (3) CORSIA, which is identical to BAU, but yearly $CO_2$ emissions are reduced by offsetting $CO_2$ emissions beyond 2020 values, (4) and (5) Flightpath 2050 (FP2050 and FP2050-cont), which utilise the targets of FP2050 (Fig. 1c, d). Note that for the CORSIA scenario, we assume an optimistic future availability and a price premium of SAF based on an analysis of feedstocks and the evolution of SAF production. As a result, approximately half (53%) of the $CO_2$ reduction that is required to achieve CORSIA's $CO_2$-neutral growth stems from the use of SAF and the other part results from carbon caps. This leads to a larger reduction in climate impact compared to a scenario where the total amount of $CO_2$ is capped. The explanation is that SAF do not only reduce the climate impact via $CO_2$ but also the reduction in contrail-cirrus climate impacts since a change in their chemical composition changes the contrail-cirrus properties (see 'Methods').

**Aviation climate impact.** We use these five scenarios to calculate their climate impact with the non-linear climate-chemistry response model AirClim[25,26] in terms of near-surface temperature change by taking into account effects from $CO_2$ as well as $NO_x$ and $H_2O$ emissions and contrail-cirrus (Fig. 2). The three

**Table 1 Summary of the scenarios used in the analysis.**

| Short Name | Long Name | Description |
|---|---|---|
| CurTec | Current Technology | Current (2012) technology is used as-is and no further political measures are implemented= 'What happens if nothing happens'= 'NoAction' |
| BAU | Business as usual | Business as usual increase in fuel efficiency without any specific aims to reduce the climate impact of aviation |
| CORSIA | Carbon-Offsetting Scheme | As BAU, a with carbon neutral growth from 2020 onwards |
| FP2050 | Flight-Path 2050 | As BAU, but including technology advancements, which are introduced according to Flightpath 2050 |
| FP2050-cont | Flight-Path 2050, continuous implementation | As FP2050, but technology advancements are introduced earlier and a smooth transition is realised |

CORSIA is the Carbon Offsetting and Reduction Scheme for international Aviation of the International Civil Aviation Organisation (ICAO), see e.g. www.icao.org.

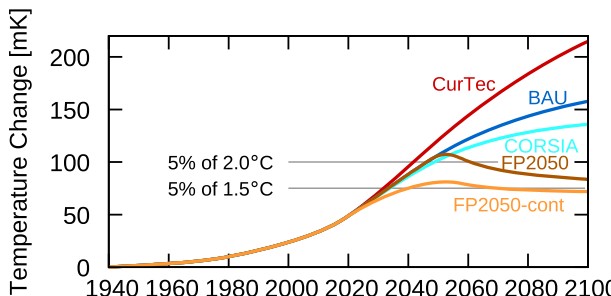

**Fig. 2 Near-surface temperature change of five scenarios including CO₂ and non-CO₂-effects.** The horizontal lines indicate 5% of a 2 °C and 1.5 °C climate target. The scenarios describe a future use of current technology, i.e. without technology improvements (CurTec, red), a business-as-usual future technological improvement (BAU, blue), the offsetting scheme of the international civil aviation organisation (CORSIA, light blue), and 2 Flightpath 2050 scenarios which differ in the speed of technology improvements (FP2050 and FP2050-cont, brown and orange, respectively).

scenarios CurTec (red line), BAU (dark blue line), and CORSIA (light blue line) show an increase in temperature until the end of the simulation (2100), though the rate of increase slows down. For CurTec, since the technology is frozen in this scenario, the rate of increase arises from the assumed development of the transport volume (Fig. 1). The increased efficiency in scenario BAU in comparison to the scenario CurTec clearly shows a substantial temperature reduction of roughly 25% in 2100. The temperature reduction is even larger for CORSIA (35–40%), due to a reduction in the effective $CO_2$ emissions from the CORSIA scheme and changes in contrail-cirrus properties from the extensive use of SAF. Terrenoire et al.[27] calculated a temperature increase in 2050 for a CORSIA scenario of 32 mK, which is consistent with our calculated value of 30.4 mK. The two implementations of the Flightpath 2050 scenarios (FP2050 and FP2050-cont) show a clear stabilisation of their climate impact, though with an overshoot around 2050. Allowing 5% of the anthropogenic temperature increase to be contributed by the aviation sector, as motivated by the current estimate of aviation to global warming, both scenarios show compliance with a 2 °C target and the scenario FP2050-cont even with the 1.5 °C target. The inertia of the climate system delays the impact of both FP2050 scenarios, which overshoot these targets around the year 2050. However, after 2050, the FP2050 measures are sufficient to cause significant temperature decreases beyond these targets from this point on.

Temperature change is a complex response to the individual measures through the various climate agents. The reductions of

the $CO_2$ emissions (Fig. 1) for all scenarios compared to the CurTec scenario lead to a significant reduction of aviation's absolute contribution to climate change (Fig. 2). However, the relative contribution to climate change, i.e. the share of $CO_2$ to the aviation's climate impact, increases from 25% in 2005 to between 33% and 56% in 2100. The reason is that the reduction in $NO_x$ emissions reduces the temperature increase via ozone faster than the reductions in $CO_2$ emissions. The short lifetime of both $NO_x$ and ozone in the atmosphere compared to $CO_2$ enables this faster response. On the other hand, the contrail-cirrus climate impact is largely driven by the distances flown. Here two factors play a role, the increase in the efficiency of the transportation system and the use of sustainable alternative fuels. These two effects lead to a reduction in the contrail-cirrus climate impact by roughly 20% in the scenarios BAU and CORSIA compared to CurTec (Fig. 3). The relative contribution of contrail-cirrus to the climate impact (Table 2) shows a reduction from 33% in 2005 to around 20% and 24% in 2100 for BAU and CORSIA scenarios, respectively, and is only slightly reduced for the FP2050 scenarios (27% and 30%). Recently, the non-$CO_2$ effects of aviation were revised concerning $NO_x$ emissions[15] and contrail-cirrus[13]. While Grewe et al.[15] stressed methodological improvements, like how to correctly attribute ozone concentrations to aviation $NO_x$ emissions, Bock and Burkhardt[13] focussed on improved contrail-cirrus microphysics. Our results include most aspects of these new developments and hence show, e.g. a larger ozone-RF as well as $NO_x$-RF compared to earlier studies, such as Lee et al.[28] (Fig. 3, left bars). The current results are in accordance with those new findings.

Hence to summarise, the increase in transport volume leads to an increase in the overall climate impact from aviation, which also increases the relative importance of $CO_2$ (25% in 2005 Base to 39% in 2100 CurTec, Table 2), even if aviation net $CO_2$ emissions are regulated and capped to 2020 values. The increase in fuel efficiency of aviation technologies at a current rate decreases the overall climate impact, especially for $CO_2$ and $NO_x$. By this, it mainly reduces the relative contribution of $NO_x$ (41–16%). The introduction of the CORSIA scheme further reduces the climate impact of $CO_2$ emissions and that increases the relative importance of contrail-cirrus and $NO_x$. The technological measures from FP2050 have a similar reduction efficiency for $CO_2$ as CORSIA, however, the strong measures for $NO_x$ largely reduce the overall climate impact so that the remaining climate impact from aviation is due to $CO_2$ (50–60%) and contrail-cirrus (around 30%).

**Contrasting aviation climate impact with 1.5 °C and 2 °C climate targets**. The climate impact of aviation emissions has a considerable uncertainty range, especially, for the non-$CO_2$ effects[22,28], which influences not only the absolute change of

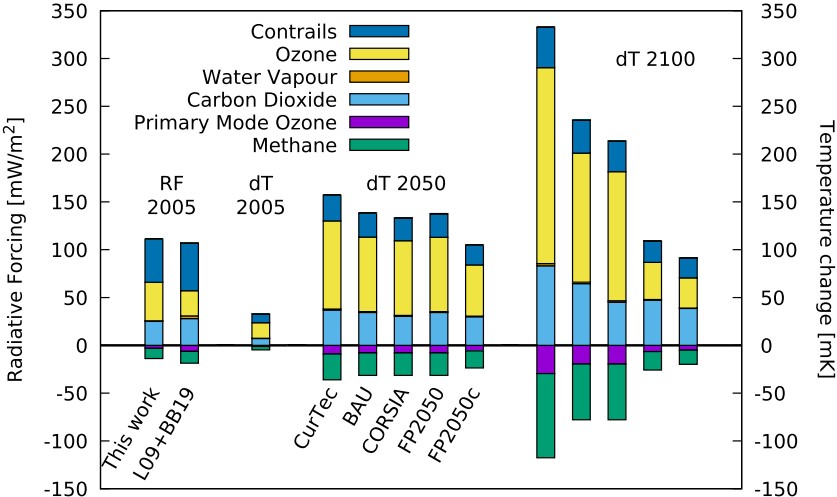

**Fig. 3 Break-down of aviation's climate impact into climate agents (colour).** The individual bars are grouped into four categories. (1) The two bars on the left describe the radiative forcing of aviation in the year 2005 (RF 2005). Results from Lee et al. (2009) are expanded by contrail-cirrus estimates based on Bock and Burkhardt (2019), denoted by L09 + BB19, respectively; (2) temperature change in the year 2005 (dT 2005); (3) temperature change in the year 2050 for the 5 scenarios (dT 2050); (4) as (3), but for the year 2100 (dT 2100). For 2100, i.e. the right-hand columns, the scenarios are presented in the same order as for 2050. The order in the legend is the same as the colours appear in the individual boxes. The scenarios describe a future use of current technology, i.e. without technology improvements (CurTec), a business-as-usual future technological improvement (BAU), the offsetting scheme of the international civil aviation organisation (CORSIA), and 2 Flightpath 2050 scenarios which differ in the speed of technology improvements (FP2050 and FP2050-cont).

**Table 2 Relative contributions of individual climate agents to climate warming (temperature change) in the year 2005 and for the 5 scenarios in 2100.**

| Contribution to warming | Year 2005 | Year 2100 | | | | | |
|---|---|---|---|---|---|---|---|
| | Base | CurTec | BAU | CORSIA | FP2050 | FP2050-cont | ECATS |
| $CO_2$ | 25% | 39% | 41% | 33% | 57% | 54% | 33–37% |
| Contrail-cirrus | 33% | 20% | 22% | 24% | 27% | 30% | 24–25% |
| $NO_x$ | 41% | 41% | 36% | 42% | 16% | 16% | 36–41% |
| $H_2O$ | 1% | 1% | 1% | 1% | 1% | 1% | 1% |

The right-hand column gives the ranges of the respective results of the ECATS bottom-up scenario. For ECATS, the minimum and maximum relative contributions out of the nine scenarios are shown and hence the minimum/maximum values do not sum up to 100%. The scenarios describe a future use of current technology, i.e. without technology improvements (CurTec), a business-as-usual future technological improvement (BAU), the offsetting scheme of the international civil aviation organisation (CORSIA), and 2 Flightpath 2050 scenarios which differ in the speed of technology improvements (FP2050 and FP2050-cont) and bottom-up estimates based on a group of experts from ECATS (Environmentally Compatible Air Transportation System).

near-surface temperatures but also the importance of the individual climate agents. In this work, we take into account uncertainties in the atmospheric lifetime of aviation-related species, uncertainties in the RF of individual species and the climate sensitivity parameter, the last of which relates the RF to temperature changes. These parameters are varied in a Monte–Carlo analysis with 10,000 simulations to obtain a range of possible atmospheric responses. The results of the Monte–Carlo analysis provide a basis for estimating a range when the temperature thresholds, 5% of 1.5 °C and 5% of 2 °C, are surpassed. For example, Fig. 4a shows the first 20 simulations of the CORSIA scenario. Figure 4b shows the probability density function (blue) and cumulative probability density function (green) of these times of surpassing the 5% of 1.5 °C for the CORSIA scenario. The mid 90% range (between the 5% and 95% percentile) indicates that this threshold is surpassed between 2025 and 2064 in the CORSIA scenario. The 5% of 2 °C is surpassed roughly 10 years later (Fig. 4c). Both, the CurTec and BAU scenario, show that both thresholds are surpassed very likely well before 2050 (Fig. 4c).

**ECATS technology scenarios and their climate impact.** The emission reductions formulated in the Flightpath 2050 are aspirational goals, which the aviation community is aiming to

achieve. Here we now contrast this with technologies which are currently discussed in the research, such as boundary layer ingestion, distributed propulsion, laminar flow control, lightweight structures, advanced geared turbofan engines, etc., and assess their potential to reduce fuel use and $NO_x$ emissions (Table 3 and Supplementary Material for more details). The majority of technology enhancements for a 2050 aircraft should, at least as an idea, be available today since the time from the development of basic research ideas (TRL 1) to having this aircraft operational in service (TRL 9) takes decades. We take into account developments for different aircraft segments, such as single-aisle and twin-aisle aircraft for entry into service between 2035 and 2050. General aviation, regional aircraft and business jet have been left out from this study, as their current contribution to total aviation $CO_2$ emission is around 5–6%, only. We take into account a large range of technologies and engine airframe integrations (see Supplementary Figs. 7, 12–16) and find a 18–22% improvement in fuel efficiency, which is similar to the analysis presented by Cumpsty et al.[2], which indicates an 18% reduction. For the far future (2050), we consider one variant for a single-aisle aircraft, while three variants are considered for a future long-range twin-aisle aircraft. These include (1) a conventional tube-and-wing wide-body aircraft (TW), (2) the so-called Flying-V

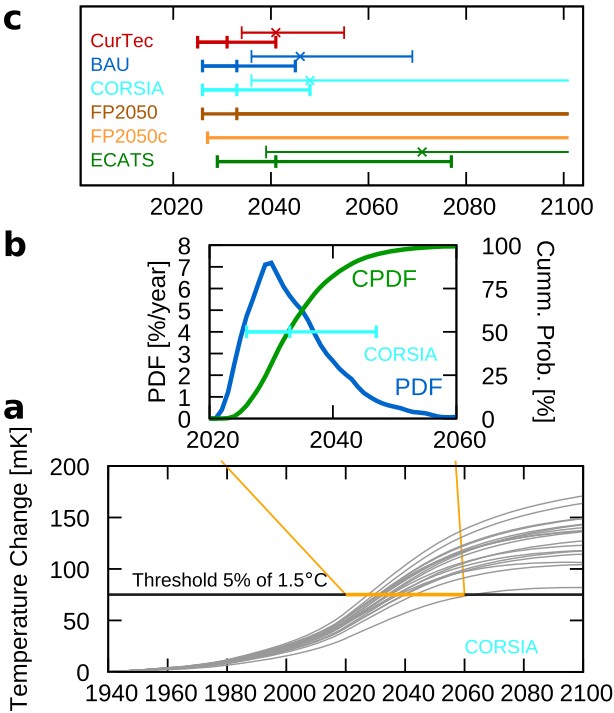

**Fig. 4 Analysis of year and likelihood that aviation surpasses a climate target. a** Potential pathways (first 20 realisations of the Monte–Carlo simulation) for the CORSIA Scenario (grey). 5% of the 1.5 °C (= 75 mK) is indicated as a black line. Crossings of the brown line with the grey line indicate the year when the threshold is surpassed. **b** Probability density function (PDF, blue) and cumulative probability density function (CPDF, green) for the year in which the climate target of 5% of 1.5 °C is surpassed (and stays above). The horizontal bar indicates the 95%, 50% and 5% percentiles. **c** 95%, 50% and 5% percentiles of the year in which the climate target is surpassed for 5% of 2 °C (thin top lines with crosses) and 5% of 1.5 °C (thick bottom lines). For both FP2050 scenarios, the 5% of 2 °C target is not surpassed in >95% of the cases, hence there is no thin line, and for FP2050 scenario with continuous improvements (FP2050-cont) the 50% percentile is beyond 2100. The scenarios describe a future use of current technology, i.e. without technology improvements (CurTec), a business-as-usual future technological improvement (BAU), the offsetting scheme of the international civil aviation organisation (CORSIA), and 2 Flightpath 2050 scenarios which differ in the speed of technology improvements (FP2050 and FP2050-cont) and bottom-up estimates based on a group of experts from ECATS (Environmentally Compatible Air Transportation System).

(FV) or multi-fuel blended wing body (MF-BWB). Both have similar aerodynamic characteristics and were developed by TU Delft[29–32] and (3) NASA's N3-X (N3) blended wing body[33,34]. We find that the fuel consumption of a 2035 aircraft might be reduced between 18% and 22% compared to new 2015 aircraft and between 34% and 44% in 2050. Note that though far future technologies, i.e. in 2075 or later, are in principle of interest they do not significantly impact our results, since their diffusion into the fleet delays their impact and more importantly, the impact on global temperatures will mainly occur beyond 2100 due to the inertia of the atmosphere-ocean system in the order of decades. These findings result in 9 ECATS emission scenarios with 3 variants (TW, FV, N3) including a pessimistic base and an optimistic implementation, which differ by ±10%. The scenarios are developed consistently with the top-down scenarios following the same transport volume development and SAF usage as in the

scenario CORSIA. Figure 5a presents the fuel use and $NO_x$ emissions relative to the year 2000, resulting in a roughly fivefold increase in fuel consumption by 2100 and a fourfold increase in $NO_x$ emission. The new technologies introduced from 2035 onwards lead to a reduction in fuel use and $NO_x$ emission around 2050, which is then offset by the further increase in transport volume, resulting in a slight increase in fuel use and $NO_x$ emission until 2100. This analysis shows that an emission pathway better than BAU might be feasible, but that the goals set by Flightpath 2050 are unlikely to be achieved. The fuel use and $NO_x$ emission from the FP2050 scenario (Fig. 1) are drastically lower than the range of our ECATS scenarios (Fig. 5).

The climate impact of the ECATS aviation scenario (Fig. 5c) shows clearly a reduction compared to the BAU scenario. However, the stabilisation of the temperature, as it was found for the Flightpath 2050 scenarios, is not achieved. The ECATS scenarios fall in between the BAU and FP2050 scenarios. The absolute change in temperature and the contribution from individual climate drivers (Table 2) contribute to climate warming in 2100 from $CO_2$ of 33–37% and the effects from non-$CO_2$ emissions roughly equally shared between contrail-cirrus and $NO_x$ emissions.

**Sensitivities to growth, global targets, sustainable fuels and technologies.** The future evolution of the aviation system and the resulting impact on climate relies on too many variables to be predicted with one outcome. To tackle this problem, we present a range of scenarios. Those are based on either an analysis of climate impacts based on set emission targets, the five scenarios mentioned in Table 1, which we call top-down scenarios, or an analysis of the climate impact of technological changes that can be expected in future aircraft, which we call the bottom-up scenarios (see 'Method'). Both approaches define possible future pathways. Even though this approach includes a large range of uncertainties, we feel that such analysis should be an important part of the debate around the impact of aviation and the potential for change within the sector. A major uncertainty is the future demand for air travel. Here we present a scenario, which lies between the estimates from Boeing and Airbus (see above) other estimates from academia[23,24] which levels off in the future. In this sense, we present a more conservative estimate of the future climate impact of aviation as compared to industry forecasts. A variation of the future growth rates by ±50% on top of the general declining growth rate leads to a change of fuel usage in the scenario BAU of roughly ±20% in 2100 and a shift in the median surpass year of 3 years (Table 4). Demand-suppressing effects from the use of more expensive SAF might end up at about 10–15% reduction of demand by 2050 for an elasticity of −1[35] and a SAF price, at best, two times that of conventional kerosene[36]. Hence, our '−50% growth rate' sensitivity simulation can be taken as an indicator for the impacts of such demand-suppressing effects, implying that the median year at which the temperature rise of 5% of 1.5 °C is surpassed will be delayed by a few years only. Most other scenarios lead to a similar shift in the median surpass year. A change in future efficiency improvements has in principle similar effects. The overall setting of the climate target and a shift from 5% to either 3.5% or 6.5% leads to a shift of the median surpass year in the order of one to two decades (Table 4). Sustainable aviation fuels are an important means in reducing the climate impact of aviation. However, according to CORSIA, whether a cap in net $CO_2$ is achieved by offsetting or the use of SAF has only a limited impact on the temperature evolution. And hence a reduction of the SAF availability by 50% leads to negligible changes in the distribution of the surpass years.

**Table 3 Estimate of potential future fuel consumption reduction with respect to A320neo (single-aisle) and A350 (twin-aisle) and estimated mission NO$_x$ improvement (last line).**

| Technology | Next Generation 2035 | | Future Generation 2050 | | | |
|---|---|---|---|---|---|---|
| | Single- aisle | Twin- aisle | Single-aisle | Twin-aisle (long-range) | | |
| | | | | Conven-tional Tube and Wing Config. | Novel aircraft | |
| | | | | | Turbofan Flying-V / MF-BWB | Turboelec. propulsionNASA N3-X |
| Airframe engine integration | +2% | +1% | −3% | −4% | −5% | −10% |
| Novel configurations | n.a. | n.a. | n.a | n.a. | −15% | −15% |
| Drag reduction | −8% | −6% | −12% | −10% | −5% | -- |
| Lightweight structures | −12% | −10% | −18% | −15% | −15% | −20% |
| Combustion based engines | −10% | −8% | −16% | −14% | −15% | −20% |
| Operations | −2% | −1% | −4% | −3% | −3% | −3% |
| Estimated fuel burn improvement | −22% | −18% | −38% | −34% | −40% | −44% |
| Estimated mission NO$_x$ improvement | −22% to −26% | −18% to −22% | −38% | −25% to −34% | −30% to −40% | −33% to −44% |

A detailed breakdown and analysis are given in the Supplementary Material. Note that the individual fuel reductions are not to be added, but multiplied (taking into account the snowball factors) to obtain the total estimated reduction. (n.a. means not applicable).

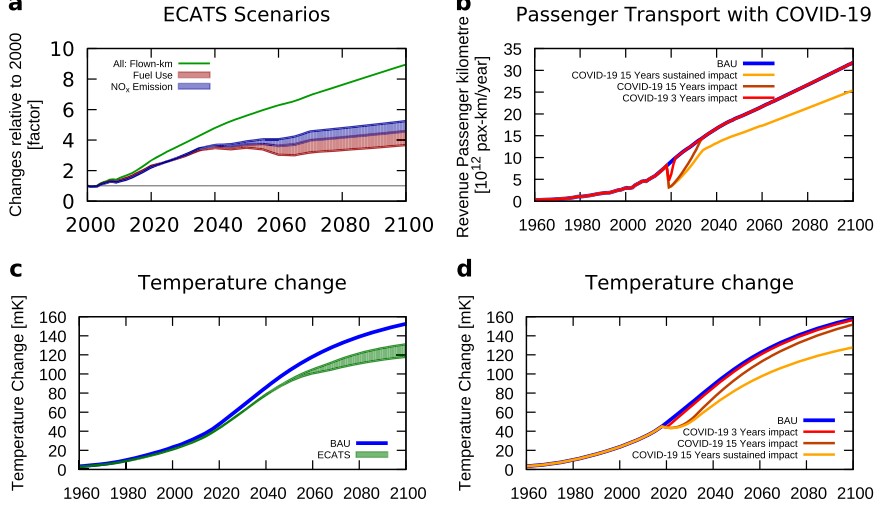

**Fig. 5 Expert judgement of feasible future technology developments (left) and the impact from the COVID-19 pandemic (right). a** as in Fig. 1, changes in fuel use (red) and NO$_x$ emissions (blue) taking into account a bottom-up analysis of aviation technologies; three far future technology pathways are taken into account, with a ± 10% uncertainty range, each leading to a scenario range; **b** Transport volume for the scenario taking into account a reduction of flight due to COVID-19 with three assumptions: a short recovery of 3 years (red); a longer recovery of 15 years (brown) and in addition to a long recovery a behavioural change after COVID-19 (yellow). **c** Resulting temperature changes as in Fig. 2 for the range of ECATS scenarios (green) and the BAU scenario for comparison (blue). **d** Resulting temperature changes from the 3 COVID-19 scenarios (red, brown, and yellow) in comparison to the BAU scenario (blue). The scenario BAU describes a business-as-usual future technological improvement and ECATS bottom-up estimates based on a group of experts from ECATS (Environmentally Compatible Air Transportation System).

**COVID-19 effects on aviation climate impact**. The recent COVID-19 pandemic might question the discussed future aviation pathways we analysed so far. To better understand the possible implications of this pandemic on the climate impact of aviation, we altered the BAU scenario in a parametric way to assess three different pathways for the international recovery from the lock-down of nation states and the associated dramatic reduction in air travel, based on reported transport volumes and scenario projections[5]. We take into account a fast recovery of 3 years, a slow recovery of 15 years (C19-03, C19-15) and a change in habits due to experiences during the lock-down, for example, a shift towards web conferences instead of face-to-face meetings. Figure 5b shows a drop in RPK due to COVID-19 and the three recovery pathways. The respective, resultant temperature change (Fig. 5d), however, is only significant if a sustained reduction in RPKs follows the crisis (yellow curve). Otherwise, the changes in 2020 due to COVID-19, as dramatic as they are for individuals and the global economy, only have a minor effect on the overall climate impact of aviation as long as a recovery follows. From the experience of other crises (e.g. SARS, 9-11, etc. see Fig. 1) we might expect a fast recovery. However, the consideration of which COVID-19 scenario is more likely is outside the scope of this study.

**Table 4 Overview on sensitivities of the surpassing time with respect to the climate target, growth rate, use of sustainable aviation fuels and fuel efficiency of new technologies.**

| 1.5 °C | Base | Target 3.5% | Target 6.5% | −50% growth | +50% growth | 50% SAF | −50% Eff. | +50% Eff. |
|---|---|---|---|---|---|---|---|---|
| CurTec | 2025 - 2031 - 2041 | 2016 - 2023 - 2033 | 2033 - 2045 - 2063 | 2026 - 2037 - 2050 | 2024 - 2033 - 2044 | n.a. | n.a. | n.a. |
| BAU | 2026 - 2033 - 2045 | 2016 - 2024 - 2035 | 2035 - 2051 - 2089 | 2027 - 2039 - 2060 | 2024 - 2034 - 2049 | n.a. | 2025 - 2036 - 2050 | 2026 - 2038 - 2059 |
| CORSIA | 2026 - 2033 - 2048 | 2016 - 2024 - 2036 | 2035 - 2058 - byd21 | 2027 - 2041 - byd21 | 2024 - 2035 - 2056 | 2025 - 2038 - 2067 | n.p. | n.p. |
| FP2050 | 2026 - 2033 - byd21 | 2016 - 2024 - 2035 | byd21 - byd21 - byd21 | 2027 - byd21 - byd21 | 2024 - 2035 - byd21 | n.p. | n.a. | n.a. |
| FP2050-cont | 2027 - byd21 - byd21 | 2016 - 2025 - 2046 | byd21 - byd21 - byd21 | byd21 - byd21 - byd21 | 2025 - 2092 - byd21 | n.p. | n.a. | n.a. |

The three years give the 5%, 50% (median), and 95% percentile. byd21 (=beyond 2100) indicates that the temperature change is not surpassing the given threshold until 2100 and might never surpass or surpass beyond 2100. (n.a. and n.p. means not applicable and simulation not performed, respectively). The scenarios describe a future use of current technology, i.e. without technology improvements (CurTec), a business-as-usual future technological improvement (BAU), the offsetting scheme of the international civil aviation organisation (CORSIA), and 2 Flightpath 2050 scenarios which differ in the speed of technology improvements (FP2050 and FP2050-cont).

## Method

**Top-down-scenario building**. In the top-down scenario building, we combine top-level assumptions on the evolution of aviation (transport volume, technologies, SAF availability) with a detailed description of the air transport system for specific years. Details are given in the Supplementary Material as textual description and EXCEL sheet. Five scenarios are assessed, which all have some common characteristics (Table 1). They have identical evolution in transport volume, defined by the revenue passenger kilometres, which resemble ICAO data for the past (1971–2017) and are extrapolated to future with the assumption of a slow decrease in traffic growth rates in future. The observed increase rate in transport volume of roughly 6% per year in the decade 2008–2017 are reduced by to roughly 1% per year in 2050 following the results from the WeCare analysis and the Randers scenario. We employed the Randers scenario named 2052 that includes the temporal development of socio-economic factors, such as population and Gross-Domestic Product, for different world regions and is complemented by reasonable narratives and scientific evaluations. Within the WeCare project, it was combined with an air passenger demand model that calculates the demand between settlements. The resulting air traffic scenario shows lower estimates of the transportation volume for the coming two decades compared to the Airbus and Boeing forecasts. The resulting air traffic scenario is not based on an extrapolation of historical trends and manufacturer expectations but considers realistic assumptions for the socio-economic growth and an associated expected saturation around 2040. Details on the forecasting methodology developed and applied in WeCare can be found in Terekhov[37] and Ghosh[38]. Future fuel efficiency improvements are based on the ICAO's environmental report[39], with 1%/year in 2018 decreasing to 0.25% in 2100. These two assumptions lead to a fuel consumption of 823 Tg in 2050, which agrees well with the mean of the ICAO scenarios[39]. The geographical distribution follows the emission inventories developed within the WeCare project[22]. Two time horizons are taken, one for the recent past (= 2012) and one representative for the future (2050), describing the geographical and vertical distribution of the emissions. All scenarios are identical between 1940 and 2018, and deviate afterwards, according to scenario assumptions, derived from the basic storylines. Thereby, we obtain 5 scenarios CurTec, BAU, CORSIA, FP2050, FP2050-cont (see main text and Table 1). The carbon-neutral growth from 2020 onwards in the CORSIA scenario is achieved by using a combination of sustainable aviation fuels (SAF) and emission offsets. Based on the EU-Renewable Energy Directive (RED-II), we assume an effective 65% net $CO_2$ reduction in SAF production and use compared to conventional kerosene in the year 2020. We assume a mix of different feedstocks, such as agricultural residues, algae, dedicated energy crops and also e-fuels (power-to-liquid), which enables an improvement of the overall $CO_2$ reduction potential to 80% in 2100. An analysis of the current growth rates and forecasts of the availability of SAF are used to optimistically estimate future availability of SAF and to allow a conservative estimate of the climate impact of the CORSIA scenario. Note that we have not explicitly considered any closed loop demand-supressing effects of increased costs[35], such as SAF costs, since EUROCONTROL has indicated that these effects might be marginal[40] and there is a high degree of uncertainty in the prediction of these costs. Instead, we have addressed this sensitivity by changing the growth rates (see below) by ±50% as open loop scenarios, which would cover a number of changes in transport volumes including those arising from demand suppressing costs increases. These assumptions lead to a scenario where 1/3 of the fuel used in 2100 is assumed to be SAF. We consider two different pathways of achieving the Flightpath 2050 objectives, late and continuous (FP2050 and FP2050-cont). Both scenarios have the same transport volume as BAU and consider technological improvements by 2050, which are formulated as '$CO_2$ emissions per passenger kilometre have been reduced by 75%, $NO_x$ emissions by 90% and perceived noise by 65%, all relative to the year 2000.'[41].

In addition to these five main scenarios, we introduce three possible development pathways related to the COVID-19 pandemic by varying the timing and degree of recovery (see main text).

**Bottom-up-scenario building**. In the Bottom-up scenario building, we present possible different development pathways and analyse how those scenarios influence the contribution of future aviation to climate change. Evolutionary technology

scenarios are developed by expert judgement (TU-Delft, Chalmers, DLR, TU-Hamburg) with comprehensive knowledge on the possible availability of advanced technologies in future aircraft programmes along with in-house tools and models for engine performance, aircraft design and aircraft performance (explained in detail within the Supplementary Material). We assess a broad spectrum of possible aircraft configurations, technologies, systems and procedures currently under research and development and evaluate their viability and provide best estimates on fuel consumption and $NO_x$ emission reduction potentials (Table 3). Comparing with the work by Schäfer et al.[42], the improvement rates are quite similar when matching our 2035 single-aisle aircraft with the evolutionary year 2035 configuration presented by Schäfer et al. The reference used in our paper is more recent and is comparable to Schäfer's 'intermediate' aircraft. They predict an 18% fuel burn reduction of the evolutionary aircraft over the intermediate aircraft, which is similar to that obtained in our analysis. In a similar approach, Hileman et al.[43] investigated at the US domestic market considering single-aisle aircraft, only. According to them, a double bubble fuselage design[44] with lower cruise speed would have 42% lower fuel consumption when compared to B737-800, which is an older generation of aircraft than the A320neo. However, it is less likely that the next generation of single-aisle aircraft will deviate from a tube and wing geometry.

In this work, the fuel efficiency and emission analysis are done for both single-aisle and twin-aisle aircraft market segments, as those two segments will account for about 95% of globally available seat kilometres. Single-aisle aircraft serving short and short-to-medium distance routes are responsible for 47% of the worldwide aviation fuel consumption. Single-and twin-aisle aircraft serving the medium and long-range routes are responsible for another 47% of the fuel consumption. Hence, differently to the top-down FP2050 scenarios, we analyse possible future technology developments and derive the expected fuel efficiencies and $NO_x$ emission evolutions in a bottom-up approach and combine that with the same overall scenario definition as for the top-down scenarios, e.g. with respect to transport volume.

We compute emission inventories based on global fleet forecast data developed in the WeCare project[22] for the years 2015–2070, in 5-year steps, for single-aisle and twin-aisle market segments. As a simplification, we assume that for each segment there is one representative aircraft type which can be used to model the entire market segment appropriately, while multiple aircraft generations are considered. The aircraft Airbus A320neo and A350 are selected as best of class for the current generation and serve as reference aircraft types for the single-aisle and twin-aisle markets, respectively. Entry into service year of the current generation is assumed to be around 2015. The next generations of single-aisle aircraft are assumed to be conventional tube-and-wing configurations entering into service in 2035 and 2050 with fuel consumption and $NO_x$ emission improvement factors as shown in Table 3 relative to the reference aircraft. For the twin-aisle market, we estimate the next generation aircraft entering into service in 2035 being a tube-and-wing configuration. In 2050, three different options, viz. a conventional tube-and-wing widebody aircraft, an aerodynamically improved aircraft, the so-called Flying-V or multi-fuel blended wing body (MF-BWB) with an advanced turbofan engine, both developed by TU Delft, and NASA's N3-X blended wing body with a turbo-electric propulsion system, are considered and used as possible twin-aisle aircraft configurations. For each of the years considered, the actual fleet composition is calculated considering a fleet diffusion of the new aircraft generations, i.e. introducing and partly replacing old aircraft. The market penetration of an aircraft generation is modelled as an S-curve applying the Bass diffusion model that has been calibrated to reach >95% market penetration within roughly 15 years, which is a typical diffusion time for new aircraft[45,46], starting from their respective entry into service (EIS) [2015, 2035, 2050].

For the calculation of the reference emission inventories (those based on the reference aircraft types), we apply the GRIDLAB methodology developed in DLR[47]. In a next step, those inventories are multiplied with the improvement factors ($CO_2$ and $H_2O$ inventories scaled according to fuel improvement, $NO_x$ inventory scaled according to $NO_x$ improvement) to determine the emissions for the respective aircraft generations. Finally, for all years, the corresponding emission inventory is obtained by combining the inventories of the individual aircraft types and generations according to their market share.

**Climate modelling**. We use the non-linear climate-chemistry response model AirClim[25,26] to analyse the climate impact of the various scenarios. AirClim is a surrogate model, which relies on a multitude of pre-calculated responses to emissions with a global climate-chemistry model and has been verified against reference models to correctly simulate scenarios, such as flying lower or higher[26]. AirClim considers changes in concentration of $CO_2$, water vapour, ozone, methane and the formation of contrail-cirrus, and takes their lifetimes, effects on the Earth radiation budget and eventually the changes in the near-surface temperature into account. The spatial resolution of the relation between emission location and response depends on the kind of effect and related atmospheric lifetimes. For $CO_2$, with a very long atmospheric perturbation, the emission location is unimportant and hence $CO_2$ concentration changes are simulated in a box model. The relation between emission location and chemical concentration changes largely depends on the altitude and geographical location of the emission. The lifetime of aviation $NO_x$ and aviation ozone is in the order of several weeks and months, respectively[48]. Accordingly, chemical responses are dependent on emission altitude and latitude, whereas for short-term contrail-cirrus effects, the longitude is also taken into account. As a background atmosphere, we take the RCP2.6 scenario into account, assuming a world which tries to achieve the Paris Agreement. The effect of sustainable aviation fuel on contrail-cirrus properties is taken into account by utilising the results from Moore et al.[49] and Burkhardt et al.[16]: A linear scaling between SAF use and reduction of soot number particle emissions is assumed, taking into account the results from measurements, which indicate that a 50–50 blend reduces the number of emitted soot particulates by 50%[49] and the change in contrail-cirrus properties and lifetime changes the contrail-cirrus RF following the results of Burkhardt et al.[16] by parameterising their results in their Fig. 1f:

$$\triangle RF^{contr} = \frac{\arctan\left(1.9 \triangle pn^{0.74}\right)}{\arctan(1.9)} \qquad (1)$$

where $\triangle RF^{contr}$ is the relative change in contrail-cirrus radiative forcing (dimensionless value between 0 and 1) and $\Delta pn$ the relative change in particle number emissions (dimensionless value between 0 and 1). Note that the formula is only valid for $\Delta pn \geq 0.1$.

The effect of SAF use on contrail-cirrus properties and lifetime changes are qualitatively in agreement with Caiazzo et al.[50]. The increase in RF when using SAF in comparison to a kerosene baseline as calculated by Caiazzo et al.[50] stems from the increase in the calculated potential contrail-cirrus coverage, which is caused in their calculations by the change in the Schmidt-Appleman criterion.

**Monte–Carlo analysis**. Uncertainties in climate impact estimates are quantified by using a Monte–Carlo Simulation. As indicated in Lee et al.[7,28] the climate impact of aviation emissions upon the atmosphere is associated with large uncertainties. The approach has been tested in Dahlmann et al.[26] and successfully applied to obtain a robust climate impact for the mitigation option Flying slower and lower[51]. Here we categorise the uncertainties into three groups following Dahlmann et al.[26]: (1) uncertainty in atmospheric residence time ($\pm 20\%$), (2) strength of RF ($\pm 5\%$ for $CO_2$, $\pm 10\%$ for $CH_4$, and $\pm 50\%$ for $H_2O$, $O_3$ (incl. PMO), and contrail-cirrus), (3) relation between RF and near-surface temperature change (climate sensitivity parameter; $\pm 5\%$ for $CO_2$, $\pm 10\%$ for $CH_4$ and contrail-cirrus, $\pm 30\%$ for $H_2O$ and $O_3$ (incl. PMO)). Hence, we consider 11 uncertainty parameters, which are drawn individually for each simulation. A total of 10,000 simulations are performed to assess the uncertainty ranges, which are displayed in Fig. 4 for the top-down scenarios. A total of 3400 simulations combined with nine different ECATS scenarios resulting in 30,600 simulations are utilised for the Monte–Carlo analysis employed in the ECATS scenarios.

## Data availability
The scenario data and result data are available on Zenodo 10.5281/zenodo.4627860.

## Code availability
The code for deriving the scenarios is given in an excel spreadsheet and available on Zenodo 10.5281/zenodo.4627860. The software code AirClim is confidential proprietary information of DLR. Therefore, the code cannot be made available to the public or the readers without any restrictions. Licensing of the code to third parties is conditioned upon the prior conclusion of a licensing agreement with DLR as licensor. The codes used for analysing the data and plotting the analysed data are available from the corresponding author upon reasonable request.

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

## Acknowledgements
The authors like to thank Dr. Christoph Kiemle for providing an internal review. The non-profit ECATS-Association IASBL (Environmentally Compatible Air Transportation System, http://www.ecats-network.eu/) promotes and supports its Members' joint activities and interests in the field of aviation and environmental impact. Its higher-level aim is to help making aviation sustainable. This study was launched and performed by ECATS members.

## Author contributions
V.G. developed the paper idea, prepared the emission data excel sheet, and performed the AirClim simulations. A.G.R., T.G., C.X., F.L. and Jo.M. analysed the top-level objectives, gave advice on how to use them in the top-down emission calculation and developed the bottom-up scenario for technical improvements. Ja.M. and B.O. analysed the legislative objectives and advised on how to use them in the top-down emission calculation. S.B., S.C. and A.G.R. analysed the effects of SAF, their potential for future use, gave advice on how to use them in the top-down emission calculation and developed the SAF part of the bottom-up scenario. K.D. and S.M. supported the AirClim simulations and interpretation.

## Funding

## Competing interests
The authors declare no competing interests.
