## [Peer Review File · Nature Communications]

Reviewer comments, first round –

Reviewer #1 (Remarks to the Author):

General comments:

- The research presented is important in principal. The linkage between aviation emission targets and the Paris agreement is a contribution.
- The modeling approach relies on system-wide evaluations of top-down and bottom-up scenarios. The main conclusions are derived by comparing both evaluations.
- System-wide evaluations require a great number of assumptions at many different levels to accommodate sources of uncertainty related to passenger demand growth, fuel price, fleet composition and turnover rate, technology introduction dates and rates, incentivizing policies, climate impact of emissions, etc.
- This study accommodated some of these uncertainties through Monte Carlo simulations; however, other sources of uncertainty were either disregarded or assumed constant, which casts doubt on the results obtained, as acknowledged by the authors themselves in lines 290 and 291 of the manuscript.

Specific comments:

- Demand growth is a huge driver in all scenarios of this study. Future growth rates are primarily derived from the WeCare framework for the near/mid-term and assumptions for the far-term. The values used deviate from the historical trend (ICAO) and are lower than those predicted by aircraft manufacturers (Boeing and Airbus). A strong justification of those values (both in the manuscript and the supplementary material) is lacking.
- Technology introduction rate is also influential for most scenarios. The top-down BAU scenario assumes a reduction in fuel use relative to the CurTec scenario using rates that were not justified. The BAU scenario was the basis for the other six top-down scenarios. Similarly, the bottom-up ECATS scenarios relied on a fleet diffusion model with coefficients of fixed values. Those values were also not justified.
- The ECATS scenario only includes two future aircraft generations introduced in 2035 and 2050. An introduction of a newer generation within the 50 years from 2050 to 2100 is highly likely and could drive the results obtained for the scenarios (could it flatten the temperature change plot of Figure 5 of the manuscript and indicate compliance with the Paris agreement?)
- The CORSIA scenario assumes a third of the fuel consumed in 2100 to be SAF. This availability assumption was mentioned to be optimistic, but not justified. Furthermore, the SAF utilized in this study was assumed to be blended with conventional jet fuels. The blended fuel would have different properties (e.g., energy content) than that of the neat SAF. It seems the neat properties were used in the equations included in the supplementary material.
- Alternative scenarios that model the aviation response to the COVID-19 pandemic were only applied to the BAU scenario. It seems the C19-15s scenario resulted in a climate impact like that of the CORSIA scenario. It would be interesting to see the impact of the COVID-19 pandemic (with a sustained reduction in air transportation volume) on the CORSIA scenario.

Other comments:

- Line 36: The word 'all' is not accurate since the referenced figure does not show sustained growth in the Northern America region.
- Line 83: It would be better to expand on the difference between the 2°C and the 1.5°C goals of the Paris agreement rather than using the 'i.e.' abbreviation.
- Figure 1: The 'net' CO₂ emission for the CORSIA scenario is plotted while the 'effective' trend is referenced in the scenario description and the figure caption. The difference between both was not immediately obvious and only became clearer after referring to the supplementary excel file. Also,

the word 'golf' was used instead of 'gulf' in the top left figure.

- Figure 2: The value of the CurTec scenario in 2100 is not shown.

- Supplementary material on top-down scenarios: Revise equations in page 6. The equation for beta does not seem correct: $[\gamma(x(i+1)) - \gamma(x(i+1))] = 0$.

Reviewer #2 (Remarks to the Author):

The manuscript "Aviation Climate Impact: global targets, the Paris Agreement and Covid-19" surely covers a topic of high relevance, so I read it with great interest. Below are my comments:

Most important concerns:

1) It took me several iterations, however, to elicit the story of the paper, as neither abstract nor the first pages really help in grasping the idea behind the paper. I would therefore encourage the authors to carefully revisit the paper to better help the reader understand the flow of the paper. My understanding of the "story" is as follows: 1) Aviation activity is expected to continue to grow over the coming decades. 2) This will lead, in the absence of mitigation measures to a further increase in the climate impact of aviation. 3) There are several industry/policy emission targets for aviation already in place, but 4) no one knows how these emission targets translate in temperature changes and 5) no one knows if it is actually feasible to achieve these targets. So the paper sets out to close the knowledge gap outlined in 4) and 5), by a comprehensive modeling of the climate impacts of achieving these emission targets and by quantifying to which degree novel technologies (aircraft, sustainable fuels) can help achieve these targets. If this is a correct reflection of the flow of the manuscript, then I would like to see this more clearly reflected in i) the abstract, ii) in the main text of the manuscript, especially, but not limited to (see also my comments further below), the first two pages of the manuscript, which in my opinion "meander" too much, and iii) also the title. Specifically, for the title I would suggest to focus more of the contrast of targets versus technologies and to drop covid-19, which is of secondary importance in the manuscript anyway (basically Covid-19 traffic scenarios serve as robustness analysis within the paper).

2) I truly appreciate the idea of contrasting the industry or political emission reduction goals with an analysis of the possibilities for climate impact reduction by the industry. However, I have two major concerns with the way that this analysis was conducted/presented. I) There have been previous papers that tried to understand what the cumulative emissions reduction production potential for aviation could be from a long list of mitigation options (including SAF, technology, ops, ATM...). See for example Hileman et al. (2013) and Schaefer et al. (2016). These papers are not cited, the contribution of the present manuscript is not delineated compared to these papers, and the findings are not contrasted. This is especially important given the fact that the Schaefer paper has a list of mitigation options considered that seems more exhaustive than the list in this manuscript here. Indeed – previous papers did not analyse the RF or temperature impact of mitigation options, but rather stopped at the level of emissions, but a significant part of the analysis chain had already done before, and one could argue at a higher resolution than in the present paper. So I would ask the authors to please position their work more clearly in the body of available literature, both with regard to contribution and to their findings. ii) I) This part of the manuscript relies on what is called in the paper as "bottom-up modeling" based on "expert assessment" or "expert judgement". There is useful supplementary material that describes the assumptions used for the modeling but there is a lack of information on how the expert assessment itself was conducted. The manuscript only mentions the institutions at which experts were working at the time of the assessment, but I am wondering about the method with which this expert assessment was done? In other words: By which process were these assumptions derived? With, for example, the Delphi Method or which method? I think these questions deserve answering in the manuscript/supplementary material.

3) Some technology options and policy measures discussed in the paper that can have a positive climate effect have a negative cost impact, in other words they drive up production costs of

airlines, so they have positive CO₂ (or RF) abatement cost. See for example the above-mentioned paper by Schaefer in Nature Climate Change for an analysis of CO₂ abatement costs. Under CORSIA, airlines have to either buy carbon offsets or use sustainable aviation fuels, both of which are associated with additional costs. If production costs of airlines increase fares will usually increase as well – and especially in a low margin business such as aviation (the low-margins are even acknowledged in the manuscript). When fares increase, travel demand decreases and aviation traffic grows more slowly (as does the climate impact). This demand-suppressing effect seems to have been unaccounted for in the analysis, both for CORSIA (see below for all CORSIA-related comments) where this omission is actually explicit from the wording in the manuscript and supplementary material, and the technology options, where one can only infer it from a lack of mentioning of costs, fares and demand.

4) The manuscript aims, among other things, to understand if the goals/policies and a portfolio of feasible technologies are sufficient to “be in line” with the Paris agreement. This is operationalized by assuming that aviation is allowed to contribute 5% of the total warming of 1.5 degrees C (or 5% of 2 degrees C), and then the probability is assessed with which certain goals/policies/technology portfolio meet this aviation-specific warming threshold. So the whole premise of the Paris angle in the paper rests upon this 5% assumption. For me it is not clear where this 5% limit comes from. The Paris agreement does not stipulate sector-specific global warming-limits. I assume, but I could be wrong, that the 5% was used in the manuscript because aviation currently contributes roughly 5% to global warming, but why is this then arbitrarily (?) defined as the maximum share of contribution for “being in line” with the Paris agreement? For me this is a substantial question mark, because the Paris agreement comparison is a core part of the manuscript.

More major comments

5) Comments specifically about the CORSIA-Modeling:

- CORSIA is, contrary what is written in the manuscript, not an emissions trading scheme. Airlines do not get/buy emission certificates that they can use or trade, but rather have to buy carbon offsets.
- CORSIA does not apply to domestic aviation, only international aviation. Has this been accounted for in the modeling?
- CORSIA is not a goal, but a policy. Flightpath is a goal. I would encourage the authors to make the distinction clearer in the manuscript.
- The CORSIA regulation contains a detailed list of carbon intensities for a set of SAF production pathways (including emissions from induced land-use change). They are part of the CORSIA package, that is, of the regulation approved by the ICAO Assembly. See ICAO (2019). The manuscript does not use this data but rather assumes a 65% emissions reduction of SAF based on the qualification threshold in the EU RED. Why has the detailed ICAO data not been leveraged in the modeling? SAF production volumes are so small (only a couple of plants are actually operating globally) that it would be feasible to combine the pathway-specific CORSIA carbon intensities with production volumes from the different plants to estimate the average carbon intensity of SAF in the year 2020.
- As mentioned above, CORSIA will increase supply costs of airlines which will lead to higher airfares and reduced demand. In line 123 it is stated that the CORSIA scenario in the manuscript assumes the Business-As-Usual assumptions BUT for CO₂ emissions and the supplementary info on the top-down modeling clarifies that traffic volumes are assumed the same between the CORSIA and the BaU scenario. So the traffic-reducing effect of higher airfares due to CORSIA has been omitted from the analysis. This is even more striking given the high uptake rates of SAF by airlines assumed in the paper and the high mark-up that airlines have to pay for SAF compared to conventional jet fuel.

6) SAF Impacts on Contrails: Line 405-415 (and 132-134: The manuscript assumes that the use of SAF reduces contrail-cirrus climate impacts. I was surprised to see that the results of Caiazza et al, 2017, in ERL, were not mentioned or discussed in the manuscript. The Caiazza paper finds that the

use of paraffinic synthetic kerosene ("SAF") leads to a net change in contrail RF from -4% to +18% among a range of assumed ice crystal habits, which seems to be in conflict with the modelling done in this manuscript here. I would appreciate an explanation.

Minor comments:

- Line 21, 22: I do not understand the meaning of this sentence. Please rephrase. But also see my general comment below on how you assessed the "being in line with Paris" question for aviation's climate targets. Did you mean that while emissions targets are ambitious enough to "be in line with Paris" the assessment of feasible technological options yields that it is unlikely that the temperature reduction associated with these targets will actually be achieved with these feasible options?

- Line 54, 55 "this number is expected to increase since aviation passenger transport has been growing at around 5% to 6% per year": Earlier in the manuscript you talk about future growth projections. Better to use those here as well, instead of historical growth, I would argue.

- Line 73-74: The Flightpath targets are per passenger-km. Please clarify in the manuscript.

- Line 109-110, Sentence "This scenario was employed within WeCare...": WeCare has not been introduced yet in the paper. Only later in the method section it becomes clear that this is the name of a project. Suggest to delete reference to WeCare here as unnecessary for story development anyway.

- Lines 298-309: Suggest to move part of the scenario description to the method section and simply say that as a robustness analysis a set of covid-19 traffic scenarios that vary by speed of recovery and degree to which face-to-face interactions by air travel is substituted by other forms of communication such as video-conferencing.

- Line 291-293 "To tackle this problem....": This is a confusing sentence, in my opinion, that I had to read several times, that goes back to the problem that I had with eliciting the story of the paper. There are simply too many different ways that "scenarios" are used in the paper and the paper would clearly benefit from more clearly distinguishing between analyses of the climate impacts of emission targets (called "top-down scenarios", which is mechanistical wording and maybe useful wording for the method section but probably not for the main text), and the climate impact of technological change ("bottom-up scenarios" - again maybe useful for the method section to explain how the modelling was done but rather meaningless for the main text). I would suggest to avoid the use of the terms bottom-up and top-down scenarios in the main manuscript.

- Line 269-274: I would like to see a quantification of increases/decreases etc. instead of the use of words such as "slight", "clear (reduction)". Here and throughout the manuscript, wherever possible.

References:

ICAO: ICAO Document 06: CORSIA Default Life Cycle Emissions Values for CORSIA Eligible Fuels , 2019.

Caiazzo, Fabio, et al. "Impact of biofuels on contrail warming." *Environmental Research Letters* 12.11 (2017): 114013.

Hileman, James I., et al. "The carbon dioxide challenge facing aviation." *Progress in Aerospace Sciences* 63 (2013): 84-95.

Schäfer, Andreas W., et al. "Costs of mitigating CO₂ emissions from passenger aircraft." *Nature Climate Change* 6.4 (2016): 412-417.

Reviewer #3 (Remarks to the Author):

This study deals with scenario analysis of emissions and global warming contributions of the aviation sector. It is an interesting and also important topic, given the fact that aviation is a carbon-intensive sector yet hard to be deeply decarbonized due to the limitations on technological improvement and aircraft replacement. Negotiations and debates on climate goal compliant actions in this industry are ongoing, which become more complicated in light of industry recovery in the post-pandemic era. This paper presents a timely study providing useful information for the relevant discussions and policy-making processes. In general, the paper is well organized and presented. Nevertheless, there are still some issues need to be addressed with regards to clarification of some contents, modelling approaches and interpretation of results. Specific comments are as follows:

- Line 49 – 71, the basic introduction of aerosol emissions and its warming mechanism is a bit too lengthy, which diminishes the main contribution of the paper. It is suggested the authors refine this part in the main context (perhaps just highlight one or two points), and put the basic knowledge in the supplementary materials.
- Scenario settings are a little bit confusing. What is the difference in terms of parameter assumptions between scenarios? Description is dispersed in the main text and the supplementary files, whereas it is difficult for the readers to explore useful information from extensive dataset. It is therefore suggested the authors may compile a table in the main text (perhaps extended from the table on Page 3 of the supplementary material "Definition of aviation top-down scenarios"), and highlight the key characteristics of each scenario.
- Line 72-75, "Among these targets is a reduction of 75% of CO₂ and 90% of NO_x emission by 2050", relative to which year or which baseline?
- In Figs. 1 and 2., there is an abrupt drop between 2050 and 2060 for "FP2050", it may need more explanation to justify the model results.
 - It seems to me that the emission reduction targets set by ACARE is used for the "FP2050" scenario, the model therefore "forced" the goal to be realized in 2050, even though the abrupt technological advancement in such as short period seems not realistic. The "FP2050C" is somehow designed to smooth this transition before 2050.
 - In this case, there should be stronger rationale for the comparison between "FP2050" and "FP2050C".
- Line 101 - 102, Page 5, caption of Fig 1. "Note that for CORSIA the effective CO₂ emission is taken into account, including reductions from the use of SAF and trading."
 - in fact, the model seems not taking into account CO₂ trading, as the data for CO₂ trading is all zeros for all the scenarios (see the CO₂ trading column in all the data sheets in the supplementary excel file). And even if it is not zero, how can you determine the volume of CO₂ being traded? Theoretically the airlines can trade all the CO₂ emissions and realize the carbon neutral target. In this case, shouldn't the emissions in the "CORSIA" scenario be zero?
- Column V in the sheet "CORSIA_optimistic" in the supplementary excel file, what is "alpha"?
- It is not clear whether SAF (Sustainable Aviation fuel) has a different emission factor of NO_x in this study, as it is a very loose definition. Different SAFs have different compositions and combustion characteristics. These assumptions have impacts on the model results that can not be ignored, and therefore should be clarified.
- The caveat of the model is that the top-down approach does not take into account the cost issue, for instance, how much cost should airlines pay for SAF adoption, emissions trading, or other technological advancement, which has significant impacts on future emissions. Aviation is an industry highly sensitive to fuel prices and other costs. Technological adoption does not typically follow a linear way, as assumed in the model. If SAF becomes cheaper someday than conventional kerosene, it might be prevalent in a relatively short time, suppose it does need too much modification of jet engines. However, other technologies such as aircraft replacement are not the case.

Response to Reviewers

General:

The authors would like to thank the reviewers for providing their time and contribution to the paper. We are encouraged by the statements of the reviewers, agreeing with us that “This paper presents a timely study providing useful information for the relevant discussions and policy-making processes. In general, the paper is well organized and presented.”, “the research presented is important in principle.” And “I truly appreciate the idea of contrasting the industry or political emission reduction goals with an analysis of the possibilities for climate impact reduction by the industry” these comments are very much in line with our motivation for preparing the paper.

The comments on the other hand were extremely helpful to identify some shortcomings, which we think have addressed in the revised version, e.g. by additional sensitivity studies, a revision of the ECATS scenario and significant textual improvements.

Your reviews have raised a number of points which we would like to address in turn below. Please find below our detailed responses (in blue and italic) to the reviewer’s comments (in black and normal font) and adaptations of the text (in blue normal font).

Reviewer 1

1. General Comments

- The research presented is important in principal. The linkage between aviation emission targets and the Paris agreement is a contribution.
- The modeling approach relies on system-wide evaluations of top-down and bottom-up scenarios. The main conclusions are derived by comparing both evaluations.
- System-wide evaluations require a great number of assumptions at many different levels to accommodate sources of uncertainty related to passenger demand growth, fuel price, fleet composition and turnover rate, technology introduction dates and rates, incentivizing policies, climate impact of emissions, etc.

Many thanks for your comments. We agree that this is an important study. It does rely on many evaluations.

2. “This study accommodated some of these uncertainties through Monte Carlo simulations; however, other sources of uncertainty were either disregarded or assumed constant”

Thank you for the comment; Indeed, there are several uncertainties which were not covered or simply discussed briefly in the discussion section. We have largely extended our analysis (23 sensitivity simulations and analysis) and varied several assumptions in order to better understand their impact on the results. The following assumptions were varied:

- *the growth rate by varying its increase rate by +/- 50%;*
- *the future efficiency improvement by varying the rate of efficiency improvement by +/- 50% in the business-as-usual scenario;*
- *the allowance of aviation to the contribution to climate change, which was set to 5% in our study, resembling current research. This parameter is now varied with a lower and upper bound of 3.5% and 6.5%.*
- *the amount of sustainable alternative fuels, used in the CORSIA: The current estimate was an optimistic view and we included a sensitivity on a scenario with only 50% SAF compared to the our optimistic CORSIA scenario.*

The results are shown in the enclosed two tables for the 1.5°C and 2.0°C climate target. For every sensitivity the 5%, 50% and 95% percentiles are given (“byd21” means that the value lies beyond 2100 or might not be met).

1.5°C	Base	Target 3.5%	Target 6.5%	-50% growth	+50% growth	50% SAF	-50% Eff.	+50% Eff.
CurTec	2025 - 2031 - 2041	2016 - 2023 - 2033	2033 - 2045 - 2063	2026 - 2037 - 2050	2024 - 2033 - 2044			
Bau	2026 - 2033 - 2045	2016 - 2024 - 2035	2035 - 2051 - 2089	2027 - 2039 - 2060	2024 - 2034 - 2049		2025 - 2036 - 2050	2026 - 2038 - 2059
CORSIA	2026 - 2033 - 2048	2016 - 2024 - 2036	2035 - 2058 - byd21	2027 - 2041 - byd21	2024 - 2035 - 2056	2025 - 2038 - 2067		
FP2050	2026 - 2033 - byd21	2016 - 2024 - 2035	byd21 - byd21 - byd21	2027 - byd21 - byd21	2024 - 2035 - byd21			
FP2050-cont	2027 - byd21 - byd21	2016 - 2025 - 2046	byd21 - byd21 - byd21	byd21 - byd21 - byd21	2025 - 2092 - byd21			

2°C	Base	Target 3.5%	Target 6.5%	50% growth	+50% growth	50% SAF	-50% Eff.	+50% Eff.
CurTec	2034 - 2041 - 2055	2023 - 2032 - 2044	2043 - 2062 - byd21	2036 - 2050 - 2075	2032 - 2043 - 2059			
Bau	2036 - 2046 - 2069	2023 - 2034 - 2049	2049 - 2092 - byd21	2038 - 2061 - byd21	2033 - 2048 - 2072		2034 - 2049 - 2074	2037 - 2060 - byd21
CORSIA	2036 - 2048 - byd21	2023 - 2035 - 2055	2052 - byd21 - byd21	2039 - byd21 - byd21	2034 - 2053 - byd21	2036 - 2063 - byd21		
FP2050	byd21 - byd21 - byd21	2023 - 2034 - byd21	byd21 - byd21 - byd21	byd21 - byd21 - byd21	2033 - byd21 - byd21			
FP2050-cont	byd21 - byd21 - byd21	2024 - byd21 - byd21	byd21 - byd21 - byd21	byd21 - byd21 - byd21	byd21 - byd21 - byd21			

Most of these parameters are dependent on cost developments, such as fuel rates. For example, fuel price influences demand and hence growth rate, or also the future fuel price will largely influence the use of sustainable alternative fuels. However, these details in economy are not simulate in detail, but are covered by the sensitivities on, e.g., growth and SAF use.

The following text has been added to the discussion.

“A variation of the future growth rates by $\pm 50\%$ on top of the general declining growth rate leads to change of fuel use in the scenario BAU of roughly $\pm 20\%$ in 2100. Additionally, a shift in the median surpass year of 2-3 years (Tab. 4) is expected while the temperature change in 2100 varies by roughly 20%. Most other scenarios lead to a similar shift in the median

surpass year. A change in future efficiency improvements has in principle similar effects. The overall setting of the climate target and a shift from 5% to either 3.5% or 6.5% leads to a shift of the median surpass year in the order of one to two decades (Tab. 4). Sustainable aviation fuels are an important means in reducing the climate impact of aviation. However, according to CORSIA, whether a cap in net CO₂ is achieved by offsetting or the use of SAF has only a limited impact on the temperature evolution. And hence a reduction of the SAF availability by 50% leads to negligible changes in the distribution of the surpass years.”

3. Demand growth is a huge driver in all scenarios of this study. Future growth rates are primarily derived from the WeCare framework for the near/mid-term and assumptions for the far-term. The values used deviate from the historical trend (ICAO) and are lower than those predicted by aircraft manufacturers (Boeing and Airbus). A strong justification of those values (both in the manuscript and the supplementary material) is lacking.

The historical demand is based on the numbers from ICAO, which is clearly stated in the text (see Figure 1) and also supplementary material; Different publication years might slightly differ wrt. transport volume data published by ICAO, however, these differences are minor, anyway. For the future development we based our scenario on the Randers scenarios. They are well described and published. More importantly they are independent from airplane manufacturers, who tend to have an optimistic view of air traffic growth.

We have adapted the text in the method section as follows:

“The observed increase rate in transport volume of roughly 6% per year in the decade 2008 to 2017 are reduced by to roughly 1% per year in 2050 following the results from the WeCare analysis and the Randers scenarios. We employed the Randers scenario ,2052’ that includes the temporal development of socio-economic factors, such as population and Gross-Domestic Product, for different world regions and is complemented by reasonable narratives and scientific evaluations. Within WeCare it was combined with an air passenger demand model that calculates the demand between settlements. The resulting air traffic scenario shows lower estimates of the transportation volume for the coming two decades compared to the Airbus and Boeing forecasts, as it is not based on an extrapolation of historical trends and manufacturer expectations but considers realistic assumptions for the socio-economic growth and an associated expected saturation around 2040. Details on the forecasting methodology developed and applied in WeCare can be found in Terekhov (2017) and Ghosh (2019).”

Future fuel efficiency improvements are based on the ICAO’s environmental report (ICAO, 2016), with 1%/year in 2018 which decreases to 0.25% in 2100. These two assumptions lead to a fuel consumption of 823 Tg in 2050, which well agrees with the mean of the ICAO scenarios (ICAO, 2016). The work by Owen and Lee (2006) and Peeters (2018) estimates a median CO₂ emission for 2050 of 2296 Mtons and 2077 Mtons, whereas our scenario is with 2586 Mtons even higher.

We have added a text passage to clarify that our values lie between other research data and the industry estimates.

In addition, the reviewer raised an important point. The growth rate is an important parameter for our calculations. Hence, we have performed 2 further Monte-Carlo simulations, varying the growth rate by +/- 50%, showing a significant shift in the time when a climate target is surpassed (see above).

4. Technology introduction rate is also influential for most scenarios. The top-down BAU scenario assumes a reduction in fuel use relative to the CurTec scenario using rates that were not justified. The BAU scenario was the basis for the other six top-down scenarios. Similarly, the bottom-up ECATS scenarios relied on a fleet diffusion model with coefficients of fixed values. Those values were also not justified.

Thank you for the comment. Indeed, the technology improvements are important and largely influence the future development in fuel use. We have based our data on the ICAO environmental report 2016. Table 1 of that report presents the technological improvement rates, which vary between 0.57%/annum to 1.5%/annum for low to optimistic assumptions. Based thereon we have chosen 1%/annum; This leads to 823 Tg fuel use in the year 2050, which compares well to the ICAO report showing around 850 Tg fuel with an uncertainty range varying roughly between 450 and 1100 Tg fuel. We have added a short comment in the method section.

The fleet diffusion in the bottom-up ECATS scenario is modelled using a generic S-curve approach (see Bass, 1969). Indeed, the coefficients have been fixed to match a typical diffusion of a new aircraft into the fleet of the respective market segment. Historical data has shown that the diffusion time (time from Entry-into-Service until nearly full market penetration is reached) is approximately 20 years. In the early age of the jet aircraft the diffusion time was only about 15 years (see also Kar et al., 2010). We have calibrated our Bass model to capture this empirical behavior knowing that radical technologies might even reduce the diffusion time further. When the diffusion curves of subsequent aircraft generations are superposed, one can observe that after a period of approximately 40 years only a very small percentage of the fleet remains in service, which is well in line with the FESG retirement curves for the single aisle and widebody aircraft from CAEP/8 (ICAO, 2008). Those coefficients have been fixed. However, the application of pessimistic and optimistic assumptions with respect to the improvement rates mimics different technology readiness levels of the assumed aircraft generations and hence simulates a shift in the EIS compared to the nominal scenario. Although this doesn't change the shape of the S-curve, it covers a possible variability in the fleet diffusion.

5. The ECATS scenario only includes two future aircraft generations introduced in 2035 and 2050. An introduction of a newer generation within the 50 years from 2050 to 2100 is highly likely and could drive the results obtained for the scenarios (could it flatten the temperature change plot of Figure 5 of the manuscript and indicate compliance with the Paris agreement?)

This is an interesting point raised by the reviewer. The two future aircraft generations are in line with the ACARE roadmap. However, looking at the historical trend of development time between two generations of aircraft, the timelines assumed in the paper could be viewed as optimistic as the likelihood of technologies assumed for 2050 might reach TRL 8 only in 2060 or later is high.

Since a considerable part of the climate impact is driven by CO₂ emissions which has a long-term cumulative effect and since the atmosphere-ocean system has a delay response time (~30 years) due to its inertia, aircraft entering after 2075 will not have significant impact on climate by 2100 and therefore the temperature change imparted by aviation till the end of this century will largely be determined by the aircraft technologies that are currently being used and those which will be introduced in the next couple of decades.

The following text has been included:

"Note that though far future technologies, i.e. in 2075 or later, are in principle of interest, they do not significantly impact our results, since their diffusion into the fleet delays their impact and more importantly the impact on global temperatures will mainly occur beyond 2100 due to the inertia of the atmosphere-ocean system in the order of decades".

6. The CORSIA scenario assumes a third of the fuel consumed in 2100 to be SAF. This availability assumption was mentioned to be optimistic, but not justified. Furthermore, the SAF utilized in this study was assumed to be blended with conventional jet fuels. The blended fuel would have different properties (e.g., energy content) than that of the neat SAF. It seems the neat properties were used in the equations included in the supplementary material.

It is designated 'optimistic' since attaining SAF market penetration has long been proven difficult to forecast and continually falls short of expectation. However, the suggested figure is consistent with estimated elsewhere in the literature.

The reviewer is correct, in that blended fuels would have properties different from their parent fuels. However, both energy content and CO₂ emissions are linearly dependent upon the blend ratio and indeed the hydrogen and carbon ratios in the final fuel blend. Hence these can be calculated separately and summed in proportion to their mass fraction.

The impact of these changes in fuel properties have been taken into account in the model.

The equations provided in the supplementary material do contain this change.

7. Alternative scenarios that model the aviation response to the COVID-19 pandemic were only applied to the BAU scenario. It seems the C19-15s scenario resulted in a climate impact like that of the CORSIA scenario. It would be interesting to see the impact of the COVID-19 pandemic (with a sustained reduction in air transportation volume) on the CORSIA scenario.

Yes indeed, there are numerous options how to combine, CORSIA with technology options, use of sustainable alternative fuels and future pathways including the effects of COVID-19. In the framework of this paper we cannot address all these aspects. Although some airlines have announced the terms of their COVID-19 bailouts, the status and details of many airline bailouts are still under discussion between airlines and regional and national governments. Local variations in operations as a result of local agreements are not taken into account, such as Air France's commitment to remove all internal flights under 2.5 hours. Reviewer 2 even suggested to drop these scenarios. So we would support leaving the COVID-19 scenarios in, but not further enlarging them.

8. Line 36: The word 'all' is not accurate since the referenced figure does not show sustained growth in the Northern America region.

Wording has been changed to "Air transport as a sector is growing rapidly in most world regions"

9. Line 83: It would be better to expand on the difference between the 2°C and the 1.5°C goals of the Paris agreement rather than using the 'i.e.' abbreviation.

The wording has been changed to better clarify this:

"Aviation is a growing sector that has committed to cap net CO₂ emissions and thus contributes to the international goals of limiting climate warming "to well below 2.0°C above preindustrial levels and pursuing efforts to limit the temperature increase to 1.5°C above preindustrial levels", as stated in the Paris Agreement (UN, 2015)."

10. Figure 1: The 'net' CO₂ emission for the CORSIA scenario is plotted while the 'effective' trend is referenced in the scenario description and the figure caption. The difference between both was not immediately obvious and only became clearer after referring to the supplementary excel file. Also, the word 'golf' was used instead of 'gulf' in the top left figure.

Wording in the Figure has been changed and caption clarified

11. Figure 2: The value of the CurTec scenario in 2100 is not shown.

The figure has been replotted to include the expected temperature rise for current Tech till 2100.

12. Supplementary material on top-down scenarios: Revise equations in page 6. The equation for beta does not seem correct: $[\gamma(x(i+1)) - \gamma(x(i+1))] = 0$.

The equation has been revised. It should read:

$\beta(k) = \gamma(k) + [\gamma(x(i+1)) - \gamma(x(i))]$...

Reviewer 2

13. It took me several iterations, however, to elicit the story of the paper, as neither abstract nor the first pages really help in grasping the idea behind the paper. I would therefore encourage the authors to carefully revisit the paper to better help the reader understand the flow of the paper. My understanding of the "story" is as follows: 1) Aviation activity is expected to continue to grow over the coming decades. 2) This will lead, in the absence of mitigation measures to a further increase in the climate impact of aviation. 3) There are several industry/policy emission targets for aviation already in place, but 4) no one knows how these emission targets translate in temperature changes and 5) no one knows if it is actually feasible to achieve these targets. So the paper sets out to close the knowledge gap outlined in 4) and 5), by a comprehensive modeling of the climate impacts of achieving these emission targets and by quantifying to which degree novel technologies (aircraft, sustainable fuels) can help achieve these targets. If this is a correct reflection of the flow of the manuscript, then I would like to see this more clearly reflected in i) the abstract, ii) in the main text of the manuscript, especially, but not limited to (see also my comments further below), the first two pages of the manuscript, which in my opinion "meander" too much, and iii) also the title.

The story outlined by the reviewer is exactly what the authors intended to convey. We have now revised the text in the introduction part in order to convey the same message in a succinct manner. Furthermore, we have made this clearer in the main text and made change to the title. Hopefully the reviewers will now find it easier to grasp the essence of the paper.

14. I truly appreciate the idea of contrasting the industry or political emission reduction goals with an analysis of the possibilities for climate impact reduction by the industry. However, I have two major concerns with the way that this analysis was conducted/presented. 1) There have been previous papers that tried to understand what the cumulative emissions reduction production potential for aviation could be from a long list of mitigation options (including

SAF, technology, ops, ATM...). See for example Hileman et al. (2013) and Schaefer et al. (2016). These papers are not cited, the contribution of the present manuscript is not delineated compared to these papers, and the findings are not contrasted. This is especially important given the fact that the Schaefer paper has a list of mitigation options considered that seems more exhaustive than the list in this manuscript here. Indeed – previous papers did not analyse the RF or temperature impact of mitigation options, but rather stopped at the level of emissions, but a significant part of the analysis chain had already done before, and one could argue at a higher resolution than in the present paper. So I would ask the authors to please position their work more clearly in the body of available literature, both with regard to contribution and to their findings

We agree that both the Hileman et al. (2013) work and the later work by Schäfer et al. (2016) have relevance for the cumulative, or bottom up, scenario study. Albeit quite relevant, we argue that both papers provide only part of the picture and we argue that the current work still has an important role to play. For two papers we make the following positioning of our work:

1. The Hileman et al. paper only addressed U.S. aviation and excluded non-CO₂ emissions from the analysis. Moreover, the technology assessment assumed that all future U.S. aviation could be modelled on a B-737 derivative, i.e. only single aisle aircraft were studied for technology updates. Although it is feasible to address U.S. aviation from this perspective, since it is quite possible to extend the range to have intercontinental operability, it is not likely that the use of twin aisle aircraft will be abandoned for the near future. Furthermore, it is known from for instance the ULTIMATE project (Grönstedt et al., 2019) that the improvement potential for the single-aisle aircraft are greater than from the twin aisle, mostly because of less improvement margin on the propulsion systems. We believe that a balanced analysis of both single and twin aisle aircraft is most consistent with the future fuel use and CO₂ emissions.

2. For the Schäfer et al. (2016) paper we agree that it is one of the best studies available when considering retrofits, discussing cost and which type of innovations that have the potential to pave its way onto future aircraft. In Schäfer's work, four mitigation categories are considered (technology, biofuels, atm and operations). Although the paper is quite exhaustive on retrofit options technology detail, we find that for the 2035 aircraft proposed in this publication only limited detail is provided (open-rotor, all-carbon fiber non-swept wing config). Still, our predictions agree quite well with the work by Schäfer and we see our work for the single aisle 2035 aircraft as a confirmation of this study. In addition, we also make a distinction between single- and twin aisle aircraft and we include technology estimates of what we expect to achieve with respect to NO_x technology to support the prediction of non-CO₂ effects.

In the light of the above arguments, the authors argue that there is a need for a bottom-up study assessing the impact of key technologies both with respect to design range (single-aisle, twin-aisle) and entry into service. This analysis should then also be related to a consistent set of predictions regarding technology for non-CO₂ emissions.

We have revised significantly our supplementary material to put our work in proper relation to the two important publications that the reviewer has pointed us to. We have also revised the results section to relate these efforts to our current effort.

15. This part of the manuscript relies on what is called in the paper as “bottom-up modeling” based on “expert assessment” or “expert judgement”. There is useful supplementary material that describes the assumptions used for the modeling but there is a lack of information on how the expert assessment itself was conducted. The manuscript only mentions the institutions at which experts were working at the time of the assessment, but I am wondering about the method with which this expert assessment was done? In other words: I think these questions deserve answering in the manuscript/supplementary material. By which process were these assumptions derived? With, for example, the Delphi Method or which method?

The supplementary material has been expanded to give more details about the assessment approach and the technologies considered in the bottom-up approach. Most of the assessment were carried out on inhouse aircraft models or those published in the literature by various authors. We have now provided a details chart about the various technologies and their expected technological improvements.

16. Some technology options and policy measures discussed in the paper that can have a positive climate effect have a negative cost impact, in other words they drive up production costs of airlines, so they have positive CO₂ (or RF) abatement cost. See for example the above-mentioned paper by Schaefer in Nature Climate Change for an analysis of CO₂ abatement costs. Under CORSIA, airlines have to either buy carbon offsets or use sustainable aviation fuels, both of which are associated with additional costs. If production costs of airlines increase fares will usually increase as well – and especially in a low margin business such as aviation (the low-margins are even acknowledged in the manuscript). When fares increase, travel demand decreases and aviation traffic grows more slowly (as does the climate impact). This demand-suppressing effect seems to have been unaccounted for in the analysis, both for CORSIA (see below for all CORSIA-related comments) where this omission is actually explicit from the wording in the manuscript and supplementary material, and the technology options, where one can only infer it from a lack of mentioning of costs, fares and demand.

This reasoning is in principle correct and requires prices and volume of offsets and alternative fuels. However: 1) The aviation CO₂ emissions are a relatively small portion of the world-wide emissions (order of 2 to 4%) 2) The CORSIA scheme covers international aviation only with a large free allowance (2019 CO₂ emissions). 3) National flights are covered within national CO₂ caps, without explicit aviation emissions caps. This implies that aviation has ample opportunities to offset excess CO₂ emissions while a relatively low price might be expected. It is expected that the price of alternative fuels is high if production volume is small, or if produced in large quantities, the price comes down quickly because the price is primarily a question of energy availability and feedstock. Simulations using the AERO-MS model (that includes stakeholders (airlines, passengers) responding to costs changes) show the impact of CORSIA, even at higher CO₂ prices, on demand is small, and on the order of only a few percent. A recent report by EUROCONTROL has confirmed this position: “It (the EUROCONTROL study) finds that there is little evidence that taxing aviation per se directly lowers CO₂ emissions; nor do raising fuel prices or ticket prices reduce CO₂ emissions. It shows that economic output is the main factor influencing demand, and hence higher or lower CO₂ emissions – and underlines that as long-distance air traffic dominates aviation emissions, efforts must be targeted on this segment if a reduction in CO₂ emissions is to be achieved.” (<https://www.eurocontrol.int/publication/does-taxing-aviation-reduce-emissions>)

The wording has been changed in the manuscript to remove ambiguity.

17. The manuscript aims, among other things, to understand if the goals/policies and a portfolio of feasible technologies are sufficient to “be in line” with the Paris agreement. This is operationalized by assuming that aviation is allowed to contribute 5% of the total warming of 1.5 degrees C (or 5% of 2 degrees C), and then the probability is assessed with which certain goals/policies/technology portfolio meet this aviation-specific warming threshold. So the whole premise of the Paris angle in the paper rests upon this 5% assumption. For me it is not clear where this 5% limit comes from. The Paris agreement does not stipulate sector-specific global warming-limits. I assume, but I could be wrong, that the 5% was used in the manuscript because aviation currently contributes roughly 5% to global warming, but why is this then arbitrarily (?) defined as the maximum share of contribution for “being in line” with the Paris agreement? For me this is a substantial question mark, because the Paris agreement comparison is a core part of the manuscript.

It is true, as mentioned by the reviewer, that there are no sector-specific global warming limits mentioned in the Paris Agreement and decisions of these limits are left to individual states. As mentioned in the World Energy Outlook of IEA, while several sectors have been successful in reducing their CO₂ emission and associated climate impact, aviation sector is one of the fastest CO₂ emission increasing sector. However, the aviation community, by and large, is committed to at least limiting its climate impact, if not decrease it. This goal has been mentioned by several organizations including ICAO, IATA, ATAG, ACARE, etc. Therefore, this paper explores the options for limiting the climate impact of aviation to its current level, which is roughly 5%.

Hence, here we are contrasting the target of limiting the temperature rise to 1.5°C and 2.0°C above pre-industrial levels with the expected climate impact from measures taken in the aviation community. We re-phrased the Paris Agreement related sentences to make this clearer.

For example, the text in the Summary:

“We show that current international political measures do reduce aviation’s climate impact, including CO₂ and non-CO₂ effects, however not at a high enough rate to stabilise aviation’s climate impact in support of the Paris Agreement”

has been revised to

“However, we also show that the reduction achieved via either of the two approaches is not sufficient enough to stabilise aviation’s climate impact in support of the overall 1.5 and 2°C climate targets of the Paris Agreement. Although we show that the emissions targets for aviation are set up in line with the overall goals of the Paris Agreement, there is a high likelihood that the climate impact of aviation will further increase without meeting these goals.”

However, the reviewer is right that the 5% target, coming from estimates of current climate impact of aviation and by this, we think. justified, not established neither in the scientific nor political debate. Therefore, we added more analysis on the dependence of our results on this choice and added a table and text in the discussion part of the paper.

18. CORSIA is, contrary what is written in the manuscript, not an emissions trading scheme. Airlines do not get/buy emission certificates that they can use or trade, but rather have to buy carbon offsets.

Reviewers statement is correct: CORSIA is an offset scheme, where CO₂ emission certificates cannot be traded.

In 2016 it was decided within the ICAO assembly that a global MBM scheme should be put in place to offset CO₂ emissions from international aviation in order to achieve carbon neutral growth. This means that CO₂ emissions beyond the 2019 level will be subjected to carbon offsetting requirements. There is no limit to the CO₂ emissions beyond the 2019 level as long as they will be offset. Also, CO₂ emissions savings below 2019 levels cannot be traded amongst entities. The amount of yearly emissions to be offset depends on the individual airline yearly emissions and a growth factor with 2019 as a reference year. The growth factor is initially (before 2030) a single global (sectoral) number that is later (beyond 2030) composed of an individual airline and sectoral contribution until after 2035 the growth factor is solely dependent on airline emissions. From 2021 to 2023 the CORSIA voluntary pilot phase will be put in action. In this period operators will annually report their fuel use on international flights within the CORSIA route network to their state authorities. Hereafter the state will inform the operator's requirements for offsetting for this three-year period. The operator must then buy and cancel emission units that are found to be eligible by ICAO.

Offsetting is different from a cap and trade system like the EU-ETS, where a maximum amount of total GHG emissions is set (the cap) and decreases over time. EU-ETS participants (e.g. airlines, power plants) that are part of the EU can trade CO₂ emissions (allowances) with other participants that are part of the EU-ETS (closed market system). CORSIA has no cap and participant cannot trade emissions amongst each other.

Wording changed throughout the manuscript to remove ambiguity and additional comments were made in the manuscript. For example in the main text page 5 (but also other places):

“The ICAO of the United Nations has agreed on a Global Market-based Measure scheme to abate the growth of CO₂ emissions from international aviation. This scheme is the “Carbon Offsetting and Reduction Scheme for International Aviation” (CORSIA). According to this scheme, the post-2020 growth in the sector must be offset such that the net carbon emissions do no longer grow. They must either be reduced via more efficient aircraft and/or the use of Sustainable Alternative Fuels (SAF) or must be compensated via offsets. CORSIA starts as a voluntary pilot scheme in 2021 and becomes mandatory, with some exceptions, in 2027 for all member states (ICAO, 2018).”

19. CORSIA does not apply to domestic aviation, only international aviation. Has this been accounted for in the modeling?

The reviewer is right that CORSIA is applied for international aviation, only. In our simplified top-down scenarios, we assess the effect of the principals of the CORSIA idea, similar to the approach of Terrenoire et al. (2019). We do not discriminate between international flights, flights in Europe and domestic flights, instead our model assumes that all flights, both international and national mimic the CORSIA scheme for aviation. See also comment 18, above.

We have adapted the text on page 5 and added a sentence to clarify this:

“Thereby we will not distinguish between domestic and international aviation but treat the sector as a whole. “

20. CORSIA is not a goal, but a policy. Flightpath is a goal. I would encourage the authors to make the distinction clearer in the manuscript.

Right, thank you for clarifying this. We absolutely share this view and adapted the text accordingly in the introduction as well as the main part of the manuscript.

21. The CORSIA regulation contains a detailed list of carbon intensities for a set of SAF production pathways (including emissions from induced land-use change). They are part of the CORSIA package, that is, of the regulation approved by the ICAO Assembly. See ICAO (2019). The manuscript does not use this data but rather assumes a 65% emissions reduction of SAF based on the qualification threshold in the EU RED. Why has the detailed ICAO data not been leveraged in the modeling? SAF production volumes are so small (only a couple of plants are actually operating globally) that it would be feasible to combine the pathway-specific CORSIA carbon intensities with production volumes from the different plants to estimate the average carbon intensity of SAF in the year 2020.

Both ICAO and RED II provide a detailed list of default LCA and ILUC values for different SAF pathway/feedstocks. Default values in ICAO embrace a significant range from net positive (e.g. HEFA/palm oil, 99.1 gCO₂/MJ) to net negative (e.g. FT/miscanthus, -22.5), with an average GHG reduction of 65% (coincidentally) across all pathway/feedstocks based on a 94.1 gCO₂/MJ fossil baseline. Whilst many of these pathways are unlikely to be commercially viable, predicting which will win out is non-trivial, particularly over the timescale of this study. However, the more fundamental point pertains to SAF production: current world SAF production is <0.1% of jet fuel usage. To increase this requires investment and construction of new plant. Within the EU all new plant must offer >65% emissions reduction to comply with RED II. Whilst many other regions do not have minimum thresholds, it is reasonable to assume that all new plant worldwide must be competitive if they wish to supply the international SAF market. In consequence 65% was deemed to be a reasonable, if slightly conservative, estimate of emissions reduction for 2025, with a linear progression to 80% by 2100. Moreover, since current world SAF production is dominated by UCO feedstocks, which are clearly finite, it would not be appropriate to combine specific plant data with ICAO defaults and extrapolate these efficiencies across the SAF sector.

22. As mentioned above, CORSIA will increase supply costs of airlines which will lead to higher airfares and reduced demand. In line 123 it is stated that the CORSIA scenario in the manuscript assumes the Business-As-Usual assumptions BUT for CO₂ emissions and the supplementary info on the top-down modeling clarifies that traffic volumes are assumed the same between the CORSIA and the BaU scenario. So the traffic-reducing effect of higher airfares due to CORSIA has been omitted from the analysis. This is even more striking given the high uptake rates of SAF by airlines assumed in the paper and the high mark-up that airlines have to pay for SAF compared to conventional jet fuel.

The reviewer is correct that CORSIA and the progressive introduction of SAF will result in an upward pressure on airfares.

Regarding CORSIA, the effect on airfares will be relatively diffuse as only flights over and above the 2019 baseline are required to offset their carbon emissions. The intention of CORSIA is to deliver net zero growth. Hence the similarity between the CORSIA and BAU scenarios is reasonable within the analysis.

Regarding SAF, this must be balanced against the opposing force of increasing pressure (both corporate responsibility and social opinion) for airlines to place sustainability more central within business models. Current driving mechanisms including offsetting (CORSIA), cap and trade (EU-ETS), mandates similar to road biodiesel (e.g. Norway 0.5% SAF), net zero corporate commitments (e.g. NIKE), government pledges (e.g. Netherlands 14% SAF by 2030), and more recently airline pledges in return for government bailouts. Whilst it undoubtedly a difficult economic decision for airlines, the direction of travel is towards increased use of drop-in SAFs as a means to radically reduce GHG emissions in aviation.

Please see also note 40.

23. SAF Impacts on Contrails: Line 405-415 (and 132-134: The manuscript assumes that the use of SAF reduces contrail-cirrus climate impacts. I was surprised to see that the results of Caiazza et al, 2017, in ERL, were not mentioned or discussed in the manuscript. The Caiazza paper finds that the use of paraffinic synthetic kerosene (“SAF”) leads to a net change in contrail RF from –4% to +18% among a range of assumed ice crystal habits, which seems to be in conflict with the modelling done in this manuscript here. I would appreciate an explanation.

Caiazza et al and the work of Burkhardt et al (2018) qualitatively agree in the estimate of changes in contrail lifetime and optical properties. The difference in the results comes from Caiazza’s calculated increase contrails “utilizing biofuels results in 8% more contrails (see their results section)”. This is a result from the higher emission index of H₂O and lower heat value - LHV (see their table 1). The slope of the mixing line of the Schmidt-Appleman Criterion (SAC) increases in their study by 8% (accidentally?). This comes from the change in quotient of EI-H₂O and LHV (1.23/43.13 vs. 1.37/44.08, see their table 1). This seems to increase the area of potential contrail coverage and by this number of contrails. In mid-latitudes, however, such a linear relationship between changes in SAC and contrails is normally not found. Contrarily, the limiting factor is the ice-supersaturation and not the SAC.

We have adapted the methods section by:

“The effect of SAF use on contrail properties and lifetime changes are qualitatively in agreement with Caiazza et al. (2017). The increase in RF when using SAF in comparison to a kerosene baseline as calculated by Caiazza et al (2017) stems from the increase in the calculated potential contrail coverage, which is caused in their calculations by the change in the Schmidt-Appleman criterion. “

Minor comments:

24. Line 21, 22: I do not understand the meaning of this sentence. Please rephrase. But also see my general comment below on how you assessed the “being in line with Paris” question for aviation’s climate targets. Did you mean that while emissions targets are ambitious enough to “be in line with Paris” the assessment of feasible technological options yields that is unlikely that the temperature reduction associated with these targets will actually be achieved with these feasible options?

Thanks for the comments. What the authors mean here is that while the emissions reduction targets are ambitious, the question is if they are in line with limiting the aviation’s climate

footprint to 5% of the 1.5°C. The introduction section has been revised, hopefully the message is conveyed better this time.

25. Line 54, 55 “this number is expected to increase since aviation passenger transport has been growing at around 5% to 6% per year”: Earlier in the manuscript you talk about future growth projections. Better to use those here as well, instead of historical growth, I would argue.

This has now been changed in the main text in accordance with the suggestions of the reviewer.

26. Line 73-74: The Flightpath targets are per passenger-km. Please clarify in the manuscript.

Text has been updated. “Among these targets is a reduction of 75% of CO₂ and 90% of NO_x emission per passenger-km by 2050. The datum for these reductions is a typical new aircraft in the year 2000.”

27. Line 109-110, Sentence “This scenario was employed within WeCare...”: WeCare has not been introduced yet in the paper. Only later in the method section it becomes clear that this is the name of a project. Suggest to delete reference to WeCare here as unnecessary for story development anyway.

The referee is right that the project itself is unnecessary for the story. However, we think that it is important to know that quite some effort has gone in to this development and the projects is also references in the method section. Hence to have this connection between the main text and the method section we rephrased the sentences including a reference.

28. Lines 298-309: Suggest to move part of the scenario description to the method section and simply say that as a robustness analysis a set of covid-19 traffic scenarios that vary by speed of recovery and degree to which face-to face- interactions by air travel is substituted by other forms of communication such as video-conferencing.

We have adapted the text according to the referee’s suggestions

29. Line 291-293 “To tackle this problem...”: This is a confusing sentence, in my opinion, that I had to read several times, that goes back to the problem that I had with eliciting the story of the paper. There are simply too many different ways that “scenarios” are used in the paper and the paper would clearly benefit from more clearly distinguishing between analyses of the climate impacts of emission targets (called “top-down scenarios”, which is mechanical wording and maybe useful wording for the method section but probably not for the main text), and the climate impact of technological change (“bottom-up scenarios” – again maybe useful for the method section to explain how the modelling was done but rather meaningless for the main text). I would suggest to avoid the use of the terms bottom-up and top-down scenarios in the main manuscript.

We have revised the text to explain better what we intend to demonstrate. We have used five possible scenarios that describe the effect of political and technical measures that have been implemented or might be implemented. However, this is a top-down approach. We also wanted to confront this approach with an approach that is based on an expert assessment of what is to be expected at the technical level. This is what we call the bottom-up approach. In the paper we confront both outcomes with each other. We have now made this clearer in the paper. We have also added an overview table of the five top-down approaches.

30. Line 269-274: I would like to see a quantification of increases/decreases etc. instead of the use of words such as “slight”, “clear (reduction)”. Here and throughout the manuscript, wherever possible.

We have now tried to avoid these subjective types of words as much as possible. With a more elaborate numerical analysis in the supplemental material and an extra overview table in the paper we have tried to quantify this as much as possible.

Reviewer 3

31. This study deals with scenario analysis of emissions and global warming contributions of the aviation sector. It is an interesting and also important topic, given the fact that aviation is a carbon-intensive sector yet hard to be deeply decarbonized due to the limitations on technological improvement and aircraft replacement. Negotiations and debates on climate goal compliant actions in this industry are ongoing, which become more complicated in light of industry recovery in the post-pandemic era. This paper presents a timely study providing useful information for the relevant discussions and policy-making processes. In general, the paper is well organized and presented. Nevertheless, there are still some issues need to be addressed with regards to clarification of some contents, modelling approaches and interpretation of results. Specific comments are as follows:

Thank you for your positive feedback!

32. Line 49 – 71, the basic introduction of aerosol emissions and its warming mechanism is a bit too lengthy, which diminishes the main contribution of the paper. It is suggested the authors refine this part in the main context (perhaps just highlight one or two points), and put the basic knowledge in the supplementary materials.

Whilst we have edited the detail to make it more readable, we do feel that it is important to include this point regarding non-CO₂ effects, including aerosol emissions and its impact as it is a central part of the context.

33. Scenario settings are a little bit confusing. What is the difference in terms of parameter assumptions between scenarios? Description is dispersed in the main text and the supplementary files, whereas it is difficult for the readers to explore useful information from extensive dataset. It is therefore suggested the authors may compile a table in the main text (perhaps extended from the table on Page 3 of the supplementary material “Definition of aviation top-down scenarios”), and highlight the key characteristics of each scenario.

See also comment 29: we have revised the text and included a new table to better illustrate the scenario set-up.

34. Line 72-75, “Among these targets is a reduction of 75% of CO₂ and 90% of NO_x emission by 2050”, relative to which year or which baseline?

The flight path 2050 goals are relative to capacities for typical new aircraft in the baseline year 2000. This is stated in the text “The datum for these reductions is a typical new aircraft in the year 2000”

35. In Figs. 1 and 2., there is an abrupt drop between 2050 and 2060 for “FP2050”, it may need more explanation to justify the model results.

From the description of the Flightpath 2050 targets, it is not totally clear when the technologies are estimated to be available. Here we span a range with a late introduction and a continuous introduction into market. We have revised the text accordingly. (see also comment above and below)

36. It seems to me that the emission reduction targets set by ACARE is used for the “FP2050” scenario, the model therefore “forced” the goal to be realized in 2050, even though the abrupt technological advancement in such a short period seems not realistic. The “FP2050C” is somehow designed to smooth this transition before 2050.
- In this case, there should be stronger rationale for the comparison between “FP2050” and “FP2050C”.

The Flightpath 2050 is not explicitly stating when the considered technologies will be ready, hence we regard here the range of possibilities. We have adapted the text to: “In the scenario “FP2050”, we consider a development of these technologies until 2050 followed by an introduction into the market, whereas in “FP2050-cont”, we use a continuous introduction of these new technologies into the market.”

37. Line 101 - 102, Page 5, caption of Fig 1. “Note that for CORSIA the effective CO₂ emission is taken into account, including reductions from the use of SAF and trading.”
- in fact, the model seems not taking into account CO₂ trading, as the data for CO₂ trading is all zeros for all the scenarios (see the CO₂ trading column in all the data sheets in the supplementary excel file). And even if it is not zero, how can you determine the volume of CO₂ being traded? Theoretically the airlines can trade all the CO₂ emissions and realize the carbon neutral target. In this case, shouldn’t the emissions in the “CORSIA” scenario be zero?

See comments at comment 16. And specific to this remark: If both national and international traffic are offsetting CO₂ emissions according to CORSIA rules, then by the aviation net CO₂ emissions equals the 2019 volume. A similar reasoning for alternative fuel holds, with a net CO₂ reduction of 70%. The offsetting is difference between the amount of CO₂ emissions to be reduced and the part which is already reduced by SAF. We have adapted the main text at various locations.

38. Column V in the sheet “CORSIA optimistic” in the supplementary excel file, what is “alpha”?

The variable is defined in the supplementary material for the top-down scenarios. The variable alpha is the ratio of CO₂ emissions as a result of the production of SAF compared to conventional fuels. This ratio is mandated to be below 65% by 2025 in the EU Renewable Energy Directive II. In this study, we envisage that this ratio will increase over the time period of this study to 80%, which represents the best available technology in 2020.

39. It is not clear whether SAF (Sustainable Aviation fuel) has a different emission factor of NO_x in this study, as it is a very loose definition. Different SAFs have different compositions and combustion characteristics. These assumptions have impacts on the model results that can not be ignored, and therefore should be clarified.

Differences in fuel composition between Jet A-1 and SAF primarily impact CO₂, H₂O, SO_x and non-volatile particulate matter emission factors. Emission factors for NO_x are equivalent. This is consistent with data in the literature.

Difference in the emission factors for fossil jet and SAF are duly considered within the model.

40. The caveat of the model is that the top-down approach does not take into account the cost issue, for instance, how much cost should airlines pay for SAF adoption, emissions trading, or other technological advancement, which has significant impacts on future emissions. Aviation is an industry highly sensitive to fuel prices and other costs. Technological adoption does not typically follow a linear way, as assumed in the model. If SAF becomes cheaper someday than conventional kerosene, it might be prevalent in a relatively short time, suppose it does need too much modification of jet engines. However, other technologies such as aircraft replacement are not the case.

Correct, the model does not consider the future cost elements such as the price of SAF. Indeed, production costs for SAF are unlikely to reach parity with production costs from mining fossil fuels. Nevertheless, corporate responsibility and social opinion are increasingly pressing the sector to place sustainability more central within their business models. Mechanisms including offsetting (CORSIA), cap and trade (EU-ETS), mandates similar to road biodiesel (e.g. Norway 0.5% SAF), net zero corporate commitments (e.g. NIKE), government pledges (e.g. Netherlands 14% SAF by 2030), and more recently airline pledges in return for government bailouts. Whilst it undoubtedly a difficult economic decision for airlines, the direction of travel is towards increased use of drop-in SAFs as a means to radically reduce GHG emissions in aviation. Note also that, here we focus on the climate impact of different assumptions in the political and technological measures. We have kept the transport volume constant to have a possibility of a clean comparison of the results. The reviewer is right that feedbacks from cost changes to the transport volume changes the picture. To cover the effects of different growth rates, we have included additional sensitivities.

Reviewer comments, second round –

Reviewer #1 (Remarks to the Author):

I have carefully reviewed the authors' response to the manuscript reviewers. The authors have sufficiently addressed all of my previous comments. I have no further comments.

Mohammed Hassan

Reviewer #2 (Remarks to the Author):

The authors conducted a thorough revision of the manuscript, which is much appreciated.

My only remaining comment pertains to the lack of accounting for demand effects as a function of the large-scale usage of SAF in the CORSIA scenario.

The authors write in their revised manuscript (lines 447-449): "We have not taken into account the effect of the increased costs of these measures. Although one might expect that cost has an influence on demand EUROCONTROL has established that these effects are marginal (EUROCONTROL, 2020)".

In the CORSIA scenario, a significant portion of the fuel used is SAF, which at minimum in the medium term will remain significantly more expensive than conventional jet fuel. Given the low-profit margins of the air transport industry, a significant part of these extra costs will be passed on to the consumers, which as long as the price elasticity of demand is non-zero will lead to a decrease in demand. The authors now argue - based on a discussion paper by Eurocontrol - that these demand effects would be marginal (compared to GDP effects, I assume), however, there is ample evidence of non-zero price elasticity of demand for air travel (see for example Morlotti et al, 2017, Brons et al., 2001, Intervistas, 2007). So my point here is that I would encourage the authors to not dismiss the demand-suppressing effect of higher costs/prices based on one reference (that, in all fairness, is not even aiming at a comprehensive quantification of the drivers of air travel demand). Instead I would suggest to simply mention the fact that the potentially demand-suppressing effect of SAF has not been accounted for in the analysis.

Morlotti, Chiara, et al. "Multi-dimensional price elasticity for leisure and business destinations in the low-cost air transport market: Evidence from easyJet." *Tourism Management* 61 (2017): 23-34.

Brons, Martijn, et al. "Price elasticities of demand for passenger air travel: a meta-analysis." *Journal of Air Transport Management* 8.3 (2002): 165-175.

Intervistas (2007): Estimating air travel demand elasticities, Final Report.

Response to Reviewers

REVIEWER COMMENTS

Reviewer #1 (Remarks to the Author):

I have carefully reviewed the authors' response to the manuscript reviewers. The authors have sufficiently addressed all of my previous comments. I have no further comments.

Mohammed Hassan

Reply: Many thanks for the positive response!

Reviewer #2 (Remarks to the Author):

The authors conducted a thorough revision of the manuscript, which is much appreciated.

My only remaining comment pertains to the lack of accounting for demand effects as a function of the large-scale usage of SAF in the CORSIA scenario.

The authors write in their revised manuscript (lines 447-449): "We have not taken into account the effect of the increased costs of these measures. Although one might expect that cost has an influence on demand EUROCONTROL has established that these effects are marginal (EUROCONTROL, 2020)".

In the CORSIA scenario, a significant portion of the fuel used is SAF, which at minimum in the medium term will remain significantly more expensive than conventional jet fuel. Given the low-profit margins of the air transport industry, a significant part of these extra costs will be passed on to the consumers, which as long as the price elasticity of demand is non-zero will lead to a decrease in demand. The authors now argue - based on a discussion paper by Eurocontrol - that these demand effects would be marginal (compared to GDP effects, I assume), however, there is ample evidence of non-zero price elasticity of demand for air travel (see for example Morlotti et al, 2017, Brons et al., 2001, Intervistas, 2007). So my point here is that I would encourage the authors to not dismiss the demand-suppressing effect of higher costs/prices based on one reference (that, in all fairness, is not even aiming at a comprehensive quantification of the drivers of air travel demand). Instead I would suggest to simply mention the fact that the potentially demand-suppressing effect of SAF has not been accounted for in the analysis.

Morlotti, Chiara, et al. "Multi-dimensional price elasticity for leisure and business destinations in the low-cost air transport market: Evidence from easyJet." *Tourism Management* 61 (2017): 23-34.

Brons, Martijn, et al. "Price elasticities of demand for passenger air travel: a meta-analysis." *Journal of Air Transport Management* 8.3 (2002): 165-175.

Intervistas (2007): Estimating air travel demand elasticities, Final Report.

Reply: Indeed, we have not included any feedback from fuel prices on demand. Our response is based on the Eurocontrol paper and we are happy to more open the discussion.

We adopted the respective passage:

“We have not taken into account the effect of the increased costs of these measures. Although one might expect that cost has an influence on demand EUROCONTROL has established that these effects are marginal (EUROCONTROL, 2020).”

The revised version reads now:

“Note that we have not considered any demand-suppressing effect of increased costs (Morlotti et al., 2017), however, EUROCONTROL has indicated that these effects also might be marginal (EUROCONTROL, 2020).”

Third round review —

Referee #2 has argued in the previous round that potential effects of demand changes need to be discussed more critically. We see that you now explicitly state that you do not discuss demand effects, but have not expanded on what evidence there is that this does not influence the results beyond the citation of the EUROCONTROL report. This reference was already given in the previous round and was not considered to be sufficient by the referee in light of other literature showing demand effects. Therefore, we would like you to discuss this aspect in more detail.

Should you be able to adequately respond to these and the reviewers' other concerns, we would be happy to look at a revised manuscript.

Reply to Reviewer #2:

We are happy to adapt the text passages as proposed by the reviewer and editor. We have explicitly stated not to have included these effects (Methods Part) and explain possible implications on the results in accordance to our previous reply. Please find below enclosed a more detailed response. We have adapted the methods and discussion section:

Methods Section:

413 ~~We have not taken into account the effect of the increased costs of these measures. Although one might~~
414 ~~expect that cost has an influence on demand EUROCONTROL has established that these effects are~~
415 ~~marginal (EUROCONTROL, 2020). Note that we have not explicitly considered any closed loop demand-~~
416 ~~suppressing effects of increased costs (Morlotti et al., 2017), such as SAF costs, since EUROCONTROL has~~
417 ~~indicated that these effects might be marginal (EUROCONTROL, 2020) and there is a high degree of~~
418 ~~uncertainty in the prediction of these costs. Instead, we have addressed this sensitivity by changing the~~
419 ~~growth rates (see below) by ±50% as open loop scenarios, which would cover a number of changes in~~
420 ~~transport volumes including those arising from demand suppressing costs increases. These assumptions~~

And in the discussion section we have added a first order estimate of this effects:

343 ±20% in 2100 and a shift in the median surpass year of 3 years (Tab. 4). Demand-suppressing effects
344 from the use of more expensive SAF might end up at about 10 to 15% reduction of demand by 2050 for
345 an elasticity of -1 (Morlotti et al. 2017) and a SAF price, at best, two times that of conventional kerosene
346 (WEF, 2020). Hence, our “-50% growth rate” sensitivity simulation can be taken as an indicator for the
347 impacts of such demand-suppressing effects, implying that the median year at which the temperature
348 rise of 5% of 1.5°C is surpassed will be delayed by a few years only. Most other scenarios lead to a

Fourth round review --

Reviewer #2 (Remarks to the Author):

My remaining comment has been appropriately addressed.